# Generalized Weighted Path Consistency for Mastering Atari Games

**Dengwei Zhao**
Shanghai Jiao Tong University
zdwccc@sjtu.edu.cn

**Shikui Tu**
Shanghai Jiao Tong University
tushikui@sjtu.edu.cn

**Lei Xu**
Shanghai Jiao Tong University
Guangdong Institute of Intelligence Science and Technology
leixu@sjtu.edu.cn

## Abstract

Reinforcement learning with the help of neural-guided search consumes huge computational resources to achieve remarkable performance. Path consistency (PC), i.e., $f$ values on one optimal path should be identical, was previously imposed on MCTS by PCZero to improve the learning efficiency of AlphaZero. Not only PCZero still lacks a theoretical support but also considers merely board games. In this paper, PCZero is generalized into GW-PCZero for real applications with non-zero immediate reward. A weighting mechanism is introduced to reduce the variance caused by scouting's uncertainty on the $f$ value estimation. For the first time, it is theoretically proved that neural-guided MCTS is guaranteed to find the optimal solution under the constraint of PC. Experiments are conducted on the Atari 100k benchmark with 26 games and GW-PCZero achieves $198\%$ mean human performance, higher than the state-of-the-art EfficientZero's $194\%$, while consuming only $25\%$ of the computational resources consumed by EfficientZero.

## 1 Introduction

In recent years, combining neural networks with search algorithms has achieved remarkable success in Reinforcement Learning (RL) [22, 24, 23]. The application scenarios include not only board games but also knowledge-free real-world problems [19]. One critical challenge is that huge computational resources are required to collect interactive experiences in order to obtain reliable policy and value networks with remarkable performance. For example, 1000 third-generation TPUs were used to generate selfplay games in MuZero [19], roughly $10 \sim 50$ years of experience per Atari game, which is unaffordable in regular situations. It is essential for an algorithm to achieve high performance with limited computational cost. Otherwise, it is difficult to generalize the algorithm to more applications.

For sequential decision problems, an interactive process between an agent and the environment is regarded as a playing path, which is collected to update the parameters of the policy and value network. In turn, the learned policy and value provide guidance for Monte Carlo Tree Search (MCTS) in determining the next action. This iterative process generates a large number of playing paths in order to obtain reliable policy and value estimators. The evaluation function $f$ at state $s_t$ is defined as the summation of $g(s_t)$, the accumulated reward from the initial state $s_0$ to $s_t$, and $h(s_t)$, the expected return received after $s_t$, which is equivalent to the state value $v(s_t)$. That is, we have

$$f(s_t) = g(s_t) + h(s_t). \tag{1}$$

To find the optimal path $L^*$ that leads to the desired termination state, the A* search algorithm [5] establishes an optimality condition for $f$ values, i.e., $f(s_t) = f(s_0)$ for all $s_t \in L^*$, where $s_0$ is

37th Conference on Neural Information Processing Systems (NeurIPS 2023).

the initial state that definitely belongs to the optimal path. In 1987, CNNEIM-A [30] relied on this optimality to improve A* by estimating the $f$-values on $L^*$ with the average of the $f$-values on the historical trajectory and on a lookahead scouting segment, plus revealing an implication that the $f$-values are regarded as i.i.d. samples from a same distribution. This idea has been adopted to equations.(8)&(9)&(10) in [29] with a regularization term added into the loss function of deep reinforcement learning such as AlphaZero [23]. The term aims at reducing the deviation of $f(s_t)$ from ones on $L^*$ towards satisfying path consistency (PC)– "*f values on one optimal path should be identical*". Simplifying equation (10) in [29] by considering $L_2$ error and ignoring weighting, recently PCZero [33] realised the PC constraint for the first time by considering

$$\mathcal{L}(\theta) = \mathcal{L}_{RL}(\theta) + \lambda \mathcal{L}_{PC}(\theta), \tag{2}$$

where $\mathcal{L}_{RL}(\theta)$ is the loss function of AlphaZero, $\lambda$ is a hyperparameter, $\mathcal{L}_{PC}$ is implemented as $(v(s_t) - \bar{v}(s_t))^2$, the squared $L_2$ deviation of the state value $v(s_t)$ from the learning target $\bar{v}(s_t)$, which is the average state value along the estimated optimal path. Experiments on three board games have demonstrated that PCZero improves the learning efficiency of AlphaZero.

However, ignoring weighting by PCZero is oversimplified because deeper states along the search path have greater uncertainty due to the exploration randomness of self-play and policy reanalysis [19]. Thus, it is essential to take into account of the uncertainty levels in $\mathcal{L}_{PC}$. Also, PCZero has only been applied to board games, wherein the immediate reward is always zero until the game terminates. Thus, $g(s_t) = 0$ holds for all states in PCZero, while PC has degenerated into the consistency of state values. It is unclear whether PC is effective generally as suggested by equation (8) in [29] to take $g(s_t)$ in consideration for applications with non-zero immediate rewards, such as Atari games. Moreover, there lacks a strong theoretical support for the effectiveness of PC, and the advantage conferred by the PC constraint has not been clearly explained.

This paper tackles the above problems by generalizing PCZero into GW-PCZero (short for **G**eneralized **W**eighted PCZero), with the main contributions summarized as follows[1].

- The application of PC is extended to scenarios where the environment emits immediate rewards, such as Atari games. Loss function in EfficientZero [32] is adopted as $\mathcal{L}_{RL}(\theta)$ in Eq. (2), instead of using the loss function of AlphaZero like PCZero [33]. $\mathcal{L}_{PC}(\theta)$ ensures the consistency of $f$ values in Eq (1) across the search path, rather than only maintaining the consistency of $h$ value like PCZero. $\bar{f}(s_t)$, the mean of the $f$ values of states along the path, is adopted as the learning target for $\mathcal{L}_{PC}$.

- An uncertainty-aware weighting mechanism is introduced for accurately estimating $\bar{f}(s_t)$. States farther away from $s_t$ are considered less reliable and given larger discounts to reduce noise in $\bar{f}(s_t)$. Experiments are conducted on the Atari $100k$ benchmark with 26 games to evaluate GW-PCZero in diverse environments. Under the same computing resource consumption, our GW-PCZero achieves $198\%$ mean human normalized performance, significantly outperforms the state-of-the-art EfficientZero's $121\%$. Additionally, GW-PCZero surpasses the original version of EfficientZero, which achieved a score of $194\%$ [32], while utilizing only $25\%$ of the computational resources required by EfficientZero.

- For the first time, it is theoretically proved that neural-guided MCTS is guaranteed to find the optimal solution under the constraint of PC, and that the optimal solution for minimizing $\mathcal{L}_{PC}(s_t)$ is achieved by satisfying the multi-step temporal difference (TD) value estimation relationship between state $s_t$ and its neighboring states in the same path. Instead of estimating state value independently, PC leads to a more reliable value estimator.

## 2   Related work

The Arcade Learning Environment (ALE) is an evaluation platform that contains plenty of Atari games, which is used to evaluate the general competency of AI agents [2]. Interaction steps are restricted to $100k$, which is roughly two hours of experience for human [14], to explore models' sample efficiency. SimPLe [8] further selects 26 games from ALE on the basis of being solvable with existing state-of-the-art reinforcement learning algorithms, which is adopted as the benchmark by many following works [9, 12, 20, 31, 13, 16]. So far, EfficientZero [32] is the only model that

---

[1]The source code is available at `https://github.com/CMACH508/GW_PCZero`.

has achieved super-human performance on Atari $100k$. Here, our generalized PC is applied to EfficientZero and improves its performance with theoretical support.

In $A^*$ search algorithm [5], the heuristic function $\hat{f}(s)$ is used to approximately estimate the optimal cost $f^*(s)$ to guide the search process. If $\hat{f}(s)$ is never over-estimated, $A^*$ search will be guaranteed to find the optimal solution. Otherwise, $A^*$ search may fail. MCTS [10, 3] is a best-first search method, which is generally not guaranteed to converge to the optimal solution. UCT [10] chose Upper Confidence Bounds (UCB) [1] as MCTS's tree policy to do simulations, and it has been proved that UCT definitely finds the optimal solution for the infinite memory case [11]. UCT evaluates states by simulating the real outcomes by fast rollouts in a Monte Carlo manner, which provides reliable information for making decisions. However, feedback value used in neural-guided MCTS in recent RL algorithms [23, 19] is provided by the predictive value network. The decision's quality highly depends on the guidance network. The optimality of MCTS is only established for part of heuristic functions and neural-guided MCTS will hardly find the optimal solution if the value estimation is poor. Though originated from $A^*$ search, PC was also promising on MCTS with deep RL algorithms like AlphaZero [29, 33]. In this paper, we prove that when constraining value estimator with PC, neural-guided MCTS used in those modern RL algorithms is guaranteed to find the optimal solution.

## 3 Preliminary

### 3.1 Path consistency

For a Markov decision process (MDP), the agent observes state $s_t$ from state space $\mathcal{S}$, chooses an action $a_{t+1}$ from action space $\mathcal{A}$, and receives reward $r_{t+1}$ according to the mapping $\mathcal{S} \times \mathcal{A} \to \mathbb{R}$. Starting from $s_0$, a playing path $L$ consisting of $n_L + 1$ states is generated while the agent interacts with the environment for $n_L$ steps:

$$L = \{s_0, s_1, s_2, \cdots, s_{n_L-1}, s_{n_L}\}. \tag{3}$$

If $s_{n_L}$ is the preferred termination state, i.e., the path $L$ receives the most accumulated reward, then $L$ becomes the optimal path $L^*$. In Eq. (1), $g(s_t)$ is the accumulated reward from $s_0$ to $s_t$ and the $f$ value can be reformulated as

$$f(s_t) = g(s_t) + v(s_t; \theta) = \sum_{i=1}^{t} r_i + v(s_t; \theta), \ \forall s_t \in L. \tag{4}$$

where $v(s_t; \theta)$ is the state value estimated by the policy-value neural network with parameter $\theta$. The general PC is defined as the consistency of $f$ values.

**Definition 3.1.** *Path consistency (PC for short) is that $f$ values of states along any optimal path in a search graph should be identical, i.e.*

$$f(s_0) = f(s_1) = f(s_2) = \cdots = f(s_{n_{L^*}}), \forall s \in L^* \tag{5}$$

### 3.2 The neural-guided Monte Carlor Tree Search (MCTS)

In MCTS, for each possible action $a \in \mathcal{A}$ in a given state $s$, a set of statistics is used to record the situation, including visit count $N(s, a)$, prior policy $p(a|s)$, cumulative reward from multiple simulations $W(s, a)$, and reward $r(s, a)$. The action value $Q(s, a)$ is calculated as $W(s, a)/N(s, a)$. Starting from the current root state $s_t$, each simulation first traverses to a leaf node $s_{t+\ell}$ by maximizing over a probabilistic upper confidence tree (PUCT) bound at each time-step, i.e.,

$$a = \arg\max_{a' \in \mathcal{A}} \left\{ \frac{W(s, a')}{N(s, a')} + p(a'|s) \frac{\sqrt{\sum_{b \in \mathcal{A}} N(s, b)}}{1 + N(s, a')} \right\}. \tag{6}$$

The children states of $s_{t+\ell}$ are expanded and $v(s_{t+\ell})$ is backed up to all edges $(s_i, a_{i+1})$ in the traversing path $\{(s_t, a_{t+1}), (s_{t+1}, a_{t+2}), \cdots, (s_{t+\ell-1}, a_{t+\ell})\}$ via

$$N(s_i, a_{i+1}) \leftarrow N(s_i, a_{i+1}) + 1,$$

$$W(s_i, a_{i+1}) \leftarrow W(s_i, a_{i+1}) + \sum_{\tau=0}^{\ell-1-i} \gamma^\tau r_{i+1+\tau} + \gamma^{\ell-i} v(s_{t+\ell}), \tag{7}$$

where $\gamma$ is the discount factor. If $\gamma$ is set to 1.0, the above equation is rewritten as:

$$W(s_i, a_{i+1}) \leftarrow W(s_i, a_{i+1}) + f(s_{t+\ell}) - g(s_i). \tag{8}$$

After $K$ time simulations, the action with the maximum visit count will be selected as the next move, and the selected child will become the new root of the search tree.

## 3.3 Reanalyze

To improve the sample efficiency, MuZero [19] proposes a reanalyze technique to revisit its past time steps. MCTS is re-executed with the latest model parameters, potentially resulting in a better-quality policy $\pi^{current}$ than the original policy $\pi^{old}$. The $l$-step bootstrapped value target is estimated based on multi-step TD value relationships

$$z_t = \sum_{j=1}^{l} \gamma^{j-1} r_{t+j} + \gamma^l v(s_{t+l}; \theta). \tag{9}$$

The reanalyze technique updates $z_t$ with the most recent value network parameters $\theta^{current}$. EfficientZero [32] employs an off-policy value correction method, which conducts an additional MCTS to estimate the value as $v^{MCTS}(s_{t+l}; \theta)$ and computes $z_t$ by replacing $v(s_{t+1}; \theta)$ with $v^{MCTS}(s_{t+l}; \theta)$ in Eq. (9). Enabling off-policy value correction for reanalysis requires running two MCTS for each sample, resulting in a doubling of the computational cost.

## 4 Methodology

The proposed GW-PCZero is built on EfficientZero with a generalized PC constraint. The PC constraint enables our GW-PCZero to not only remove the time-consuming off-policy value correction which has a significant contribution to the performance of EfficientZero, but also improve the performance to surpass EfficientZero with theoretical guarantee on the optimality. The loss function for learning the policy-value network is taken as the same form of Eq. (2). Different from PCZero, here $\mathcal{L}_{RL}(\theta)$ is set to be the same as EfficientZero's loss function, consisting of policy loss, value loss, value prefix loss, and self-supervised consistency loss [32], while $\mathcal{L}_{PC}(\theta)$ is evaluated by the deviation of $f$ value from its mean along the path. Specifically, on the state $s_t$, we can rewrite the squared $L_2$ deviation $[f(s_t) - \bar{f}(s_t)]^2$ by noticing $v(s_t; \theta) = h(s_t)$ in Eq. (1) as follows:

$$\mathcal{L}_{PC}(s_t; \theta) = \left\{ v(s_t; \theta) - [\bar{f}(s_t) - g(s_t)] \right\}^2. \tag{10}$$

Built on the basis of MuZero, EfficientZero has achieved superhuman performance on Atari games [32] with limited $100k$ training data. EfficientZero has made three critical changes to MuZero. First, a self-supervised consistency loss was proposed to help model the environment's state transition dynamics. Second, one-step reward prediction was replaced by the estimation of value prefix $\sum_{j=1}^{l} \gamma^{j-1} r_{t+j}$. Third, MCTS value target correction was executed by replacing the policy-value network's output with root values estimated by MCTS. To take advantage of the merits of EfficientZero, we keep the first two changes and disable the third one, because the MCTS's root value target correction doubles the need of computing resources. Experimental results indicate that values estimated by the network trained under the PC constraint is more reliable than this time-consuming value correction method. With the help of the PC constraint, our method is able to improve the efficiency of the policy-value network parameter learning and save computing resources.

### 4.1 The optimality of the PC loss

It is noted from Eq. (10) that $\mathcal{L}_{PC}(s_t)$ is the square error loss and $\mathcal{L}_{PC}(s_t) \geq 0$ holds. We present the following theorem for the optimality of the PC loss.

**Theorem 4.1.** *The PC loss of state $s_t$ achieves the optimal value when the following equation holds:*

$$\begin{cases} v(s_t) = \displaystyle\sum_{j=t+1}^{i} r_j + v(s_i), & i > t, \\ v(s_i) = \displaystyle\sum_{j=i+1}^{t} r_j + v(s_t), & i < t. \end{cases} \tag{11}$$

*The above Eq. (11) actually represents the multi-step TD value estimation relationships between $s_t$ and all other states $\{s_i\}_{i \neq t}$ in the same path.*

The estimated values given by an effective value function predictor should satisfy the multi-step TD value relationships specified by the Bellman equation. Theorem 4.1 indicates that $\mathcal{L}_{PC}$ will be minimized if multi-step TD value relationships are established for the estimated values, indicating that path consistency is a nature that a well-learned value predictor should have. The term of PC loss guides the learning process towards it proactively, which gives an explanation about why the adoption of PC improves learning efficiency. State values learned by minimizing $\mathcal{L}_{PC}$ will establish the relationship among them, rather than learning the value predictions for different states independently. In particular, by setting discount factor $\gamma = 1$ in Eq. (9), the bootstrapping target for the state $s_i$'s value learning in MuZero becomes the right formula of Eq. (11), implying that PC is beneficial to the learning of value estimation. It should be noted that Theorem 4.1 is also established for any $L_p$ norm deviation loss with $p \geq 1$, not limited to $L_2$ norm used in this paper.

To prove Theorem 4.1, we define $t_{PC}(s_t) = \bar{f}(s_t) - g(s_t)$ as the learning target for the value network in Eq. (10), and let $\bar{f}(s_t)$ be the mean over all states in the path $L$. Based on Eq. (3) & (4), $\mathcal{L}_{PC}(s_t; \theta)$ is equivalently derived to be

$$
\mathcal{L}_{PC}(s_t; \theta) = \frac{1}{(n_L + 1)^2} \left\{ \sum_{i > t} \left[ v(s_t; \theta) - \left( \sum_{j=t+1}^{i} r_j + v(s_i) \right) \right] \right.
$$
$$
\left. - \sum_{i < t} \left[ v(s_i) - \left( \sum_{j=i+1}^{t} r_j + v(s_t; \theta) \right) \right] \right\}^2 . \tag{12}
$$

According to Eq. (12), the PC loss is a square error function with $\mathcal{L}_{PC}(s_t; \theta) \geq 0$. When the multi-step TD value relationships by Eq. (11) are satisfied, each term in the square brackets of Eq. (12) is equal to zero. Then, $\mathcal{L}_{PC}(s_t; \theta)$ is minimized at zero, and it implies that PC is established, which suggests that Theorem 4.1 holds. The derivation details of Eq. (12) are given in Appendix 1.

Previous research extensively employed TD error for training value function models in various algorithms. Minimizing TD error is a commonly utilized learning objective among model-free RL algorithms, such as Q learning [28] and its variants [15, 27, 6, 26]. For model-based RL algorithms, the TD error is employed as one of the auxiliary training objective during the learning process of the environment model. Dyna [25] updated the estimated values to be consistent with the environment model by minimizing the TD(0) error, while TD($k$) is adopted by value equivalent models, such as MuZero [19] and Muesli [7]. SCZero [4] utilizes the TD(0) error to update the environment model and value function simultaneously to encourage them to be jointly self-consistent, demonstrating that the TD constraint on value estimations is beneficial for both policy evaluation and environment model construction. Compared with those TD based algorithms, PC is a global constraint for all states on the optimal path. As illustrated in Eq. (12), PC loss of $s_t$ is minimized if the TD errors for state $s_t$ with all states on the same path are minimized to be 0, while TD error itself represents a local constraint that pertains to the relationship between two states. The PC constraint on state $s_t$ encompasses both consistency with subsequent states and consistency with preceding states. In contrast, the TD error only takes into account the consistency with subsequent states. Both SCZero and our GW-PCZero propose a consistency constraint to improve the learning efficiency. SCZero [4] is built on model-based algorithms, encouraging the learned environment model and value function to be consistent by minimizing the TD(0) error. The concept of path consistency, explored in this paper, is conceptually different from the notion of self-consistency discussed in SCZero. PC requires that estimated $f$ values on the same optimal path are consistent, which is applicable to all value-based RL algorithms, regardless of their reliance on a model-based framework.

We further prove that the PC constraint on the estimated values guarantee the neural-guided MCTS to find the optimal solution, under an assumption below, which is made from the perspective of probability based on Definition 3.1. We use the notation $\mathcal{N}(\mu, \sigma^2)$ to denote a normal distribution and use $Pr(\mu, \sigma^2)$ to denote an arbitrary distribution, where $\mu$ denotes mean and $\sigma^2$ denotes variance.

**Assumption 4.2.** *For the state $s$ on the optimal path, suppose $f(s) = \mu_0^f$; For every state $s'$ not on the optimal path, $\{f(s')\}$ are assumed to be sampled from an i.i.d. $Pr(\mu_1^f, \sigma_1^2)$.*

The $\mu_0^f$ and $\mu_1^f$ are the expected total rewards for optimal and non-optimal solutions respectively according to the definition of $f$, and the inequality $\mu_0^f > \mu_1^f$ holds.

**Theorem 4.3.** *Under the PC condition, $P_g$, defined as the probability of finding the optimal path with the neural-guided MCTS, satisfies the following inequality as the number of simulations $K$ goes sufficiently large,*

$$\lim_{K \to +\infty} P_g \geq \lim_{K \to +\infty} \prod_{t=1}^{n_L} \left\{ 1 - \frac{1}{2} \exp \left\{ -\frac{[b_t(\mu_0^f - \mu_1^f)]^2}{2\sigma_1^2/K'} \right\} \right\}^{m-1}, \qquad (13)$$

*where $b_t$ is a constant, and $K'$ is the minimum simulation times for nodes in the search tree.*

The proof of Theorem 4.3 is given in Appendix 2. Intuitively, if $K$ goes large ($K \to +\infty$), every child in the search tree will be visited enough times because of the exploration term in Eq. (6), leading to $K' \to +\infty$. Then, the lower bound of Eq. (13) approaches one, which pushes the probability of finding the optimal path to be one. Formally, we have the following result.

**Theorem 4.4.** *For any Markov decision process, the neural-guided MCTS with path consistency as the constraint of the estimated values is guaranteed to find the optimal solution, when the number of simulations $K$ is sufficiently large,*

$$\lim_{K \to +\infty} P_g = 1. \qquad (14)$$

## 4.2 Empirical computation of PC on $f$ consistency

Considering that interaction sequences in practical applications like Atari games are quite long, the generated samples are used to update parameters before reaching the termination. In practice, the state $s_t$ and its $\ell$ following unrolled states are sampled from the replay buffer as a small training batch $L_b = \{s_t, s_{t+1}, \cdots, s_{t+\ell}\}$. Reanalyzing is adopted to re-estimate the learning target with the latest policy-value network for all states in $L_b$. States in $L_b$ share part of the cumulative reward before $s_t$. The PC target $t_{PC}$ for the value network in Eq. (10) is calculated as

$$t_{PC}(s_t) = \frac{1}{l+1} \sum_{i=0}^{l} \left[ \sum_{j=1}^{i} r_{t+j} + v(s_{t+i}; \theta) \right]. \qquad (15)$$

Only $s_t$ is considered to be constrained with PC loss in real implementation because the sampled batch $L_b$ is too short to deal with the subsequent states in the same way. The reanalyze technique is used to prepare the learning targets for policy and value without enabling off-policy value correction. All target preparations are summarized in Algorithm 1.

---

**Algorithm 1:** Sample Preparation for GW-PCZero

**Input**: Replay buffer $\mathcal{R}$, Unrolled steps $l$, Batch size $N$.
**Output**: $(\pi, z, t_{PC})$.
  1: Sample $N$ unrolled sequences from $\mathcal{R}$ and each sequence consists of $l+1$ consecutive states.
  2: **for** each sampled sequence $L_i = \{s_t^i, s_{t+1}^i, \cdots, s_{t+l}^i\}$ **do**
  3:     Reanalyze policy target $\pi$ by performing one iteration MCTS for each state in $L_i$.
  4:     Recalculate value target $z$ by bootstrapping in Eq. (9) for each state in $L_i$.
  5:     Estimate PC target $t_{PC}(s_t)$ according to Algorithm 2.
  6: **end for**
  7: **return** Tuple $(\pi, z, t_{PC})$.

---

## 4.3 Weighting the PC target

The estimation of $t_{PC}(s_t)$ in Eq. (15) takes the mean of all $f$ values, treating all states equally. However, the states far away from $s_t$ are less reliable than the ones near $s_t$, which is mainly caused by two operations, as shown in Figure 1. One is the exploration uncertainty during the iteration process with the environment. As $s_{t+i}$ is sampled from the policy distribution injected with noise, it may not be the optimal choice. The other more important reason is policy reanalyzing. As illustrated in

the right of Figure 1, $s_{t+1}$ is the following state of $s_t$ in the real iteration sequence collected with an older policy $\pi^{old}$, but the optimal choice of the latest reanalyzed policy $\pi^{current}$ may be another state $s_{t+1}^*$. According to Definition 3.1, PC requires that all states should be on the same optimal path, and it will be violated due to the above two operations. Especially, the probability that $s_{t+i}$ and $s_t$ are not in the same optimal path grows as $i$ increases.

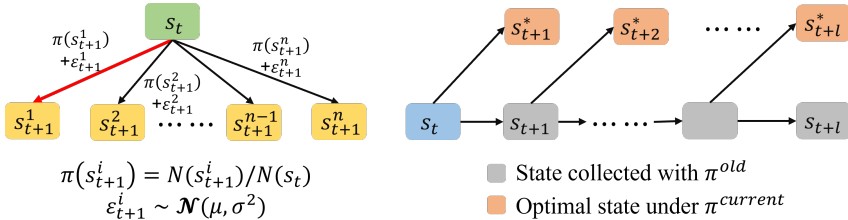

Figure 1: Left: exploration uncertainty when playing the game. Action is sampled from $\pi$ and $\varepsilon$ is a Gaussian noise. Right: difference between collected sequence $\{s_{t+i}\}$ and current reanalyzed policy's optimal choice $\{s_{t+i}^*\}$.

In order to mitigate the high variance brought by the above uncertainty, a weighting mechanism is devised to give a larger discount to farther states. Specifically, $t_{PC}(s_t)$ is calculated by a weighted summation and the weight $w_i$ decreases linearly with index $i$, i.e.,

$$w_i = \max\{c_b - c_a \times i, 0\}, \tag{16}$$

where $c_a$ and $c_b$ are two constants. While there are numerous algorithms available for weighting, the linear weighting method employed in this paper is simple and highly interpretable, which deserves further investigation in future research. The PC target $t_{PC}$ is calculated as:

$$t_{PC}(s_t) = \sum_{i=0}^{l} \widetilde{w}_i \left[ \sum_{j=1}^{i} r_{t+j} + v(s_{t+i;\theta}) \right], \tag{17}$$

where $\widetilde{w}_i = w_i / \sum_{i=0}^{l} w_i$ is the normalized weight. The details are summarized in Algorithm 2.

---

**Algorithm 2:** Weighted PC target $t_{PC}$ estimation

---

**Input**: $S = \{s_t, r_{t+1}, s_{t+1}, \cdots, s_{t+l}\}$, value function $v(s)$.
**Output**: Target $t_{PC}(s_t)$.
1: Initialize the value summation $T \leftarrow 0$ and weight summation $W \leftarrow 0$.
2: **for** each state $s_{t+i}$ in S **do**
3:   Calculate weight $w_i$ for state $s_{t+i}$ according to Eq. (16).
4:   $T \leftarrow T + w_i \times \left[ \sum_{j=1}^{i} r_{t+j} + v(s_{t+i}) \right], W \leftarrow W + w_i$
5: **end for**
6: **return** target $t_{PC}(s_t) \leftarrow T/W$.

---

### 4.4 The coefficient for the PC loss

PC is incorporated into the policy-network learning as a regularization term in Eq. (2), and there is a non-negative coefficient $\lambda$ to be determined. GW-PCZero degenerates back to a variant of EfficientZero (without MCTS value target correction) when $\lambda = 0$. In general, an appropriate $\lambda$ is required to balance between maximizing the accumulated reward and satisfying the PC constraint. According to the empirical experience of PCZero on Hex game [33], it has been found that PC improves the generalization ability for testing while sacrificing the prediction accuracy for training. For games with low complexity, it is easy to generate all possible states for training and $\lambda$ should be small to improve the training accuracy. For games with high complexity, the generalization ability should be paid more attention by giving a larger $\lambda$. See the details in Appendix 3.

Therefore, $\lambda$ is adjusted with the game complexity, which is evaluated by two criteria. One is the action space size $|\mathcal{A}|$, which determines the number of brunch factors for tree search. The other is the difficulty for the computer program to master, which is approximately evaluated by $D_m =$ *MuZero Score/Human Score* in Atari games, where MuZero is trained without $100k$ limitation. In the experiment, if $|\mathcal{A}| < 18$, $\lambda$ is set to be 0.2. For games with $|\mathcal{A}| \geq 18$, if $D_m$ is larger than 100, indicating that this game is easier to master, $\lambda$ will be set to be 0.35. Otherwise, this game is difficult to master, and $\lambda = 0.4$.

## 5 Experiments

Experiments are conducted on Atari $100k$ benchmark containing 26 games [8]. Only $100k$ interaction steps are allowed to be collected, i.e., $400k$ frames based on a frameskip of 4, which is roughly the number of experiences played by a human in two hours. The mean performance of human players was collected [17], which is used to evaluate the model's performance against the baseline of the random player's score. The human normalized score of a learned agent is defined as $(Score_{agent} - Score_{random})/(Score_{human} - Score_{random})$.

Table 1: Comparison of different algorithms. The computational cost is measured by the number of MCTS executions.

| | Value correction | Weighting | Adjustable $c_a$ | Frames | Train steps | Cost | Score |
|---|---|---|---|---|---|---|---|
| EfficientZero | ✓ | N/A | N/A | $100k$ | $120k$ | $61.54M$ | 1.943 |
| EfficientZero[†] | × | N/A | N/A | $100k$ | $60k$ | $15.46M$ | 1.218 |
| G-PCZero | × | × | × | $100k$ | $60k$ | $15.46M$ | 1.378 |
| GW-PCZero | × | ✓ | × | $100k$ | $60k$ | $15.46M$ | 1.980 |
| GW-PCZero($c_a$) | × | ✓ | ✓ | $100k$ | $60k$ | $15.46M$ | 2.066 |

Table 2: Scores on the Atari $100k$ benchmark tested with 32 evaluation seeds. Benchmark methods include SimPLE [8], CURL[12], DrQ [31], and SPR+resets [16]. EfficientZero[†] [32] is retrained under the same computational resources as GW-PCZero.

| Game | Human | SimPLe | CURL | DrQ | SPR+resets | EfficientZero[†] | G-PCZero | GW-PCZero |
|---|---|---|---|---|---|---|---|---|
| Alien | 7127.7 | 616.9 | 558.2 | 771.2 | **911.2** | 850.6 | 761.3 | 699.7 |
| Amidar | 1719.5 | 88.0 | 142.1 | 102.8 | **201.7** | 60.6 | 91.6 | 97.0 |
| Assault | 742.0 | 527.2 | 600.6 | 452.4 | 953.0 | 994.8 | 1093.3 | **1224.1** |
| Asterix | 8503.3 | 1128.3 | 734.5 | 603.5 | 1005.8 | **17734.4** | 15712.5 | 14771.9 |
| BankHeist | 753.1 | 34.2 | 131.6 | 168.9 | **547.0** | 276.9 | 280.9 | 207.2 |
| BattleZone | 37187.5 | 5184.4 | 14870.0 | 12954.0 | 8821.2 | **15875.0** | 13875.0 | 13500.0 |
| Boxing | 12.1 | 9.1 | 1.2 | 6.0 | 32.2 | 28.2 | 11.8 | **41.6** |
| Breakout | 30.5 | 16.4 | 4.9 | 16.1 | 23.4 | 366.7 | 402.7 | **450.0** |
| ChopperCmd | 7387.8 | 1246.9 | 1058.5 | 780.3 | **1380.6** | 818.8 | 1221.9 | 1150.0 |
| CrazyClimber | 35829.4 | 62583.6 | 12146.5 | 20516.5 | **28936.2** | 8059.4 | 8031.3 | 9734.4 |
| DemonAttack | 1971.0 | 208.1 | 817.6 | 1113.4 | 2778.0 | 7940.8 | 15163.4 | **24074.1** |
| Freeway | 29.6 | 20.3 | **26.7** | 9.8 | 18.0 | 0.0 | 0.0 | 0.0 |
| Frostbite | 4334.7 | 254.7 | 1181.3 | 331.1 | **1834.3** | 229.1 | 245.6 | 249.7 |
| Gopher | 2412.5 | 771.0 | 669.3 | 636.3 | 930.4 | **1325.6** | 851.3 | 1286.9 |
| Hero | 30826.4 | 2656.6 | 6279.3 | 3736.3 | 6735.6 | 7537.2 | **9958.1** | 8171.3 |
| Jamesbond | 302.8 | 125.3 | 471.0 | 236.0 | 415.7 | 300.0 | 298.4 | **525.0** |
| Kangaroo | 3035.0 | 323.1 | 872.5 | 940.6 | **2190.6** | 525.0 | 1012.5 | 262.5 |
| Krull | 2665.5 | 4539.9 | 4229.6 | 4018.1 | 4772.4 | 3818.5 | 4233.0 | **7782.0** |
| KungFuMaster | 22736.3 | 17527.2 | 14307.8 | 9111.0 | 14682.1 | 8956.3 | 10059.4 | **20543.8** |
| MsPacman | 6951.6 | 1480.0 | 1465.5 | 960.5 | 1324.6 | 967.5 | 784.7 | **1594.1** |
| Pong | 14.6 | 12.8 | $-16.5$ | $-8.5$ | $-9.0$ | 15.6 | **19.8** | 19.8 |
| PrivateEye | 69571.3 | 58.3 | **218.4** | $-13.6$ | 82.2 | 0.0 | 100.0 | 96.9 |
| Qbert | 13455.0 | 1288.8 | 1042.4 | 854.4 | 3955.3 | 8120.3 | 8281.3 | **13651.6** |
| RoadRunner | 7845.0 | 5640.6 | 5661.0 | 8895.1 | 13088.2 | 3443.7 | 9262.5 | **16809.4** |
| Seaquest | 42054.7 | 683.3 | 384.5 | 301.2 | 655.6 | 478.1 | **838.1** | 768.1 |
| UpNDown | 11693.2 | 3350.3 | 2955.2 | 3180.8 | **30185.0** | 7592.5 | 6129.4 | 12344.7 |
| Normed Mean | 1.000 | 0.443 | 0.381 | 0.357 | 0.901 | 1.212 | 1.378 | **1.980** |
| # >Human | 0 | 2 | 2 | 2 | 7 | 7 | 7 | **11** |

Table 3: Scores on the Atari $100k$ benchmark with different $c_a$ in Eq. 16. $c_a = 0$ means no weighting. $c_a = 0.1$ has been used in Table 2.

| $c_a$ | Seaquest | Asterix | Boxing | Alien | BattleZone |
|---|---|---|---|---|---|
| 0.0 | 838.1 | 15712.5 | 11.8 | 761.3 | 13875.0 |
| 0.05 | **1215.6** | **22131.3** | 22.0 | 568.1 | 14281.3 |
| 0.1 | 768.1 | 14771.9 | 41.6 | 699.7 | 13500.0 |
| 0.2 | 776.9 | 14950.0 | **53.8** | **982.5** | **16812.5** |

| $c_a$ | Breakout | Qbert | KungFuMaster | Krull | Kangaroo |
|---|---|---|---|---|---|
| 0.0 | 402.7 | 8281.3 | 10059.4 | 4233.0 | **1012.5** |
| 0.05 | 443.9 | 12568.8 | 12575.0 | 6874.9 | 787.5 |
| 0.1 | **450.0** | **13651.6** | **20543.8** | **7782.0** | 262.5 |
| 0.2 | 396.7 | 11269.5 | 10990.6 | 2182.5 | 206.3 |

In EfficientZero [32], MCTS consumes the majority of computing resources, much more than network updating. The number of MCTS is determined by two factors: data collection, which requires $100k$ MCTS for $100k$ steps, and data reanalysis. $120k$ updating steps are performed with a batch size of 256. A total of 30.72 million samples are reanalyzed, with each sample requiring two MCTS runs: one for policy re-estimation and the other for MCTS value correction. Thus, a total of 61.44 million MCTS runs are used to prepare the training data, and the computational cost for data generation is relatively negligible. Therefore, the limit of $100k$ interaction steps does not fully constrain the computational resources used, and the total number of MCTS runs should be considered. Due to the reanalysis operation, EfficientZero requires a total of 61.54 million MCTS runs, which accounts for the majority of the computational resources consumed.

In this paper, we evaluate our GW-PCZero's learning efficiency under a more difficult setting than the original EfficientZero [32]. As summarized in Table 1, the updating steps are limited to $60k$, and MCTS off-policy value correction is disabled. Models in our experiment only consume $25\%$ computational resources of the original EfficientZero. G-PCZero is included for comparisons by using the unweighted computation of the PC target in Eq. (15). EfficientZero[†] is retrained under identical conditions. Experiments are conducted on 4 NVIDIA Tesla A100 GPUs with 16 CPU cores. We set $c_b = 1.0$ and $c_a = 0.1$ in Eq. (16). Totally 32 of different random seeds are used. Other hyperparameter settings are the same as EfficientZero, as summarized in Appendix 5.

The mean testing results are reported in Table 2. GW-PCZero performs better than EfficientZero[†] in 19 out of all 26 Atari games. The mean of the human normalized score of GW-PCZero achieves $198.0\%$, much higher than EfficientZero[†]'s $121.2\%$. Moreover, GW-PCZero slightly outperforms the original EfficientZero [32], which achieved a score of $194.3\%$. It should be noted that GW-PCZero only consumes $25\%$ computational resources of the original EfficientZero, as shown in Table 1. Experiment results demonstrate the effectiveness of our GW-PCZero, and indicate that PC improves the learning efficiency remarkably in Atari games. Notice that GW-PCZero performs poorly in several games requiring great exploratory, such as Freeway, probably due to the quick convergence of PC. Moreover, examples of learning curves of GW-PCZero and EfficientZero[*] on Atari games are displayed in Appendix 6. Analysis of test variances is included in Appendix 4.

GW-PCZero outperforms G-PCZero in 15 games and achieves a significantly higher mean normalized score. This ablation study indicates that the weighting mechanism is important to provide a more accurate target for PC learning. The performance is greatly improved by considering different uncertainty levels in the states along the path while calculating the mean $f$ values. It is critical to observe that G-PCZero outperforms EfficientZero[†] in 16 Atari games, and its mean score of $137.8\%$ is slightly higher than that of EfficientZero[†], indicating that PC improves the model's performance even without the weighting mechanism. We also employ the weighting mechanism in PCZero [33] and conduct experiments on board games. The game performance exhibits improvement as well. Details are included in Appendix 7.

Path consistency requires that $f$ values on the optimal path should be identical. The selection of the optimal path from the collected interaction experiences is not only necessary but also presents a significant challenge. Due to the difficulty of definitively determining the optimality of a particular path, one alternative approach is to strive for the collection of training paths that are near-optimal, thus

satisfying the requirement of PC. In on-policy reinforcement learning, game experiences collected through interactions between the updated policy and the environment improve gradually and approach a near-optimal level throughout the training process. Alternatively, near-optimal paths obtained from human experts or other high-level computer players can also fulfill the requirement of PC, allowing the model to be trained in an off-policy manner. Both aforementioned training approaches have been demonstrated to be effective in PCZero [33]. The critical factor for the success of PC lies in the selection of the optimal path, and is relatively unaffected by whether the path is collected through on-policy or off-policy methods. However, in many scenarios, the availability of near-optimal paths provided by experts is limited, and on-policy learning methods demonstrate low sample efficiency. Therefore, in addition to the aforementioned methods, it is crucial to investigate the training of PC using suboptimal paths. For example, game frames in Atari $100k$ are collected using a poorly trained model in the early state, which are far from near-optimal. GW-PCZero implements a weighting mechanism to address the limitation of suboptimal paths by excluding unreliable nodes along the collected path from consideration, and experimental results in Table 2 demonstrate the effectiveness of this node filtering approach.

We further investigate the weighting mechanism in more detail. As shown in Eq. (16), linear decay is adopted from the initial value $c_b = 1.0$, with the decay rate controlled by $c_a$. A larger $c_a$ gives less weight to the latter states, while $c_a = 0$ represents the situation without weighting. The effects of $c_a$ are illustrated in Table 3. The optimal $c_a$ varies across different games, indicating that there is still room for improving the weighting strategy. On one hand, more reliable neighboring states should be considered to reduce the variance of the mean $f$ value. On the other hand, unreliable neighbors increase uncertainty in the estimation of the mean $f$ value. $c_a$ is to achieve a trade-off between including more neighbors and reducing unreliability. Notice that when considering Asterix, Alien, BattleZone, and Kangaroo, GW-PCZero is inferior to EfficientZero in Table 2, but it becomes better than EfficientZero under appropriate values of $c_a$. When $c_a$ is adjustable, GW-PCZero($c_a$) achieves a mean score of 2.066, with only 2 out of 26 games performing worse than EfficientZero[†]. These results suggest that there is potential for further improvement in the weighting strategy and that more effort should be devoted to exploring this area in future research.

## 6   Conclusion

In this paper, we have proposed GW-PCZero to consider path consistency (PC) in a general setting and provided a theoretical analysis of the convergence property of MCTS with the PC constraint. GW-PCZero extends the previous PCZero from board games to more practical applications, for which the environment emits immediate rewards, such as Atari games. An uncertainty-aware weighting mechanism, which gives a larger discount to the farther states along the searching path, is devised to compute a more reliable learning target for PC. What's more, we have theoretically proved that MCTS is guaranteed to find the optimal solution under the constraint of PC. Experiments demonstrate that PC is beneficial to obtain strong artificial intelligence programs but consumes fewer computational resources. Neural-guided MCTS has been extensively utilized to address optimization problems in various practical applications, such as de novo drug molecular generation [18], organic molecule retrosynthesis [21] and so on. There is potential for the development of PC to contribute to addressing these critical problems by providing more robust value estimators. This work is still in the nascent stages without further applications related to people's daily lives currently and thus there is no immediate ethical or harmful social impacts.

## 7   Acknowledgement

This work was supported by the National Natural Science Foundation of China (62172273), and Shanghai Municipal Science and Technology Major Project (2021SHZDZX0102). Shikui Tu and Lei Xu are corresponding authors.

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
