# 1 Deviation of Eq (12)

For a generated path containing $n_L + 1$ states, $L = \{s_0, s_1, \cdots, s_{n_L}\}$, $v(s_t)$ is the estimated state value, and $z(s_t) = \sum_{j=t+1}^{n_L} r_j$ is the ground truth for value estimation by setting discount factor $\gamma = 1$. PC loss's learning target for state $s_t$ is $\bar{f}(s_t)$, the mean of $f$ values along the path, minus $g(s_t)$. $\bar{f}(s_t)$ is calculated as Equation 1.

$$
\begin{aligned}
\bar{f}(s_t) &= \frac{1}{n_L + 1} \sum_{i=0}^{n_L} f(s_i) = \frac{1}{n_L + 1} \sum_{i=0}^{n_L} \left( \sum_{j=1}^{i} r_j + v(s_i) \right) \\
&= \frac{1}{n_L + 1} \left[ \sum_{i=0}^{n_L} v(s_i) + \sum_{i=0}^{n_L} \sum_{j=1}^{i} r_i \right] = \frac{1}{n_L + 1} \left[ \sum_{i=0}^{n_L} v(s_i) + \sum_{i=1}^{n_L} (n_L + 1 - i) r_i \right] \\
&= \sum_{i=1}^{n_L} r_i + \frac{\sum_{i=0}^{n_L} v(s_i) - \sum_{i=1}^{n_L} i \times r_i}{n_L + 1} = \sum_{i=1}^{t} r_i + \sum_{i=t+1}^{n_L} r_i + \frac{\sum_{i=0}^{n_L} v(s_i) - \sum_{i=1}^{n_L} \sum_{j=i}^{n_L} r_i}{n_L + 1} \\
&= g(s_t) + z(s_t) + \frac{\sum_{i=0}^{n_L} [v(s_i) - z(s_i)]}{n_L + 1}
\end{aligned}
\tag{1}
$$

Therefore, PC loss is

$$
\begin{aligned}
\mathcal{L}_{PC}(s_t) &= [v(s_t) - (\bar{f}(s_t) - g(s_t))]^2 \\
&= \left\{ v(s_t) - z(s_t) - \frac{\sum_{i=0}^{n_L} [v(s_i) - z(s_i)]}{n_L + 1} \right\}^2 \\
&= \frac{\left\{ \sum_{i \neq t} [v(s_t) - z(s_t) - v(s_i) + z(s_i)] \right\}^2}{(n_L + 1)^2} \\
&= \frac{1}{(n_L + 1)^2} \left\{ \sum_{i > t} \left[ v(s_t; \theta) - \left( \sum_{j=t+1}^{i} r_j + v(s_i) \right) \right] \right. \\
&\quad \left. - \sum_{i < t} \left[ v(s_i) - \left( \sum_{j=i+1}^{t} r_j + v(s_t; \theta) \right) \right] \right\}^2.
\end{aligned}
\tag{2}
$$

Eq (13) is derived.

# 2 Deviation of Theorem 4.3

To prove Theorem 4.3, we first introduce three lemmas.

**Lemma 2.1.** *Assume $x \sim \mathcal{N}(\mu_1, \sigma_1^2)$, $y \sim \mathcal{N}(\mu_2, \sigma_2^2)$, where $\mathcal{N}(\cdot, \cdot)$ denotes normal distribution. If $x$, $y$ are independent of each other and $\mu_2 > \mu_1$, then*

$$
P(x > y) = \frac{1}{2} \exp \left\{ -\frac{1}{2} \frac{[(\mu_2 - \mu_1)/\sqrt{\sigma_1^2 + \sigma_2^2}]^2}{\cos^2 \xi} \right\}
\tag{3}
$$

*where $0 < \xi < \pi/2$ is a constant.*

**Lemma 2.2.** *Assume $x_i \sim \mathcal{N}(\mu_i, \sigma_i^2), \forall i = 1, 2, \cdots, m$. Variables in $\{x_i\}$ are independent of each other. $\mu_1 > \mu_i, \forall i = 2, 3, \cdots, m$. Then*

$$
P[x_1 = \max\{x_1, x_2, \cdots, x_m\}] = \prod_{i=2}^{m} \left\{ 1 - \frac{1}{2} \exp \left\{ -\frac{(\mu_1 - \mu_i)^2}{2(\sigma_1^2 + \sigma_i^2) \cos^2 \xi_i} \right\} \right\}
\tag{4}
$$

15   Proof of Lemma 2.1 was previously given in [4], and the details are summarized as follows.

16   $\because f(x, y) = f(x)f(y)$, to see Fig 1 (a), we have

$$P(x > y) = \iint\limits_{x>y} f(x)f(y)dxdy$$

$$= \iint\limits_{x>y} \frac{1}{\sqrt{2\pi}\sigma_0} \exp\left\{-\frac{1}{2}\left(\frac{x-\mu_0}{\sigma_0}\right)^2\right\} \times \frac{1}{\sqrt{2\pi}\sigma_1} \exp\left\{-\frac{1}{2}\left(\frac{y-\mu_1}{\sigma_1}\right)^2\right\} dxdy \qquad (5)$$

17   Let $(x - \mu_0)/(\sqrt{2}\sigma_0) = u$, $(y - \mu_1)/(\sqrt{2}\sigma_1) = v$, then $|J| = 2\sigma_0\sigma_1$, where $|J|$ is the Jacobian
18   determinant.

19   Domain $D$: $x > y$ become $D1$: $\sqrt{2}\sigma_0 + \mu_0 > \sqrt{2}\sigma_1 v + \mu_1$.

20   By formula

$$\iint\limits_{D} f(x, y)dxdy = \iint\limits_{D_1} f[x(u, v), y(u, v)]|J|dudv \qquad (6)$$

$$P(x > y) = \frac{1}{\pi} \iint\limits_{D_1} \exp[-(u^2 + v^2)]dudv$$

$$= \frac{1}{\pi} \iint\limits_{D_1+D_2} \exp[-(u^2 + v^2)]dudv - \frac{1}{\pi} \iint\limits_{D_2} \exp[-(u^2 + v^2)]dudv \qquad (7)$$

21   See Fig. 1 (b), let $u = \rho\cos\phi$, $v = \rho\sin\phi$.

$$\iint\limits_{D_1+D_2} \exp[-(u^2 + v^2)]dudv = \int_{-\pi+\phi_1}^{\phi_1} d\phi \int_0^\infty \exp(-\rho^2)d\rho = \pi/2 \qquad (8)$$

22   where $r = [(\mu_1 - \mu_0)/(\sqrt{2}\sigma_0)] \times [\sin\phi_1/\sin(\phi_1 - \phi)]$.

$$\iint\limits_{D_2} \exp[-(u^2 + v^2)]dudv = \int_{-\pi+\phi_1}^{\phi_1} \left[\int_0^r \rho\exp(-\rho^2)d\rho\right] d\phi$$

$$= \frac{\pi}{2} - \frac{1}{2}\int_{-\pi+\phi_1}^{\phi_1} \exp\left\{-\left(\frac{(\mu_1 - \mu_0) \times \sin\phi_1}{\sqrt{2}\sigma_0 \times \sin(\phi_1 - \phi)}\right)^2\right\} d\phi$$

$$= \frac{\pi}{2} - \frac{1}{2}\int_{-\pi/2}^{\pi/2} \exp\left\{-\left(\frac{(\mu_1 - \mu_0) \times \sin\phi_1}{\sqrt{2}\sigma_0 \times \cos\theta}\right)^2\right\} d\theta \qquad (9)$$

23   Let $\theta = \phi_1 - \phi - \pi/2$. Therefore,

$$P(x > y) = \frac{1}{\pi}\int_0^{\frac{\pi}{2}} \exp\left\{-\left(\frac{\mu_1 - \mu_0}{\sqrt{2}\sigma_0}\sin\phi_1\right)^2 / \cos^2\theta\right\} d\theta$$

$$= \frac{1}{\pi}\int_0^{\frac{\pi}{2}} \exp\left\{-\frac{1}{2}\left(\frac{\mu_1 - \mu_0}{\cos\theta\sqrt{\sigma_0^2 + \sigma_1^2}}\right)^2\right\} d\theta \qquad (10)$$

24   $\because \arctan\phi_1 = \sigma_0/\sigma_1$, $\therefore \sin\phi_1 = \sigma_0/\sqrt{\sigma_0^2 + \sigma_1^2}$. From the Mean-value theorem for integrals [1]

$$P(x > y) = \frac{1}{\pi}\int_0^{\frac{\pi}{2}} \exp\left\{-\frac{1}{2}\left(\frac{\mu_1 - \mu_0}{\sqrt{\sigma_0^2 + \sigma_1^2}}\right)^2 / \cos^2\theta\right\} d\theta$$

$$= \frac{1}{2}\exp\left\{-\frac{1}{2}\left(\frac{\mu_1 - \mu_0}{\sqrt{\sigma_0^2 + \sigma_1^2}}\right)^2 / \cos^2\xi\right\} \qquad (11)$$

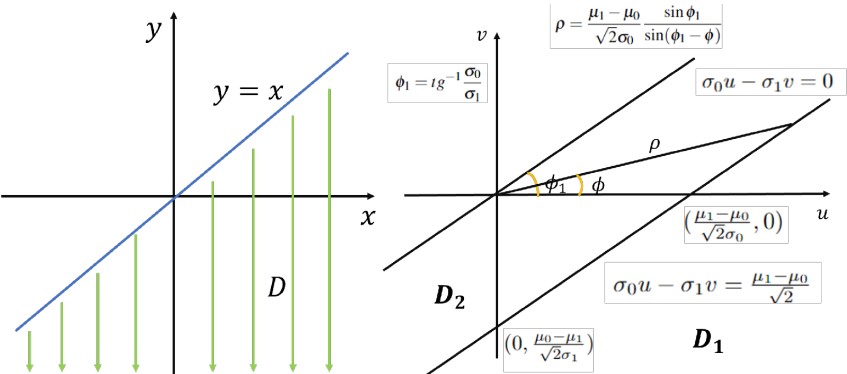

Figure 1: Left: (a), Right: (b)

25 Where $0 < \xi < \pi/2$ is a constant. Q.E.D.

26 Lemma 2.2 is proved according to Eq. (3) on the basis of Lemma 2.1, as follows:

$$P[x_1 = \max\{x_1, x_2, \cdots, x_m\}] = \prod_{i=2}^{m} P(x_1 \geq x_i)$$

$$= \prod_{i=2}^{m} [1 - P(x_i > x_1)]$$

$$= \prod_{i=2}^{m} \left\{ 1 - \frac{1}{2} \exp \left\{ -\frac{(\mu_1 - \mu_i)^2}{2(\sigma_1^2 + \sigma_i^2) \cos^2 \xi_i} \right\} \right\}. \tag{12}$$

27 **Lemma 2.3.** *Lindeberg–Lévy Central Limit Theory [2]: Suppose $\{X_1, \cdots, X_n\}$ is a sequence of*
28 *i.i.d. random variables with $E[X_i] = \mu$ and $Var[X_i] = \sigma^2 < \infty$. Then*

$$\lim_{n \to \infty} \bar{X}_n \sim \mathcal{N}(\mu, \sigma^2/n). \tag{13}$$

29 Next, we give the proof of the Theorem 4.3. While doing MCTS, a scouted subtree $T(s_t)$ is generated
30 and it contains $K + 1$ nodes including $s_t$ after $K$ simulations as illustrated in the left of Figure 2.
31 Based on Eq. (8), the estimated root state value $v(s_t)$ is calculated as:

$$v(s_t) = \frac{\sum_{s' \in T(s_t)} f(s')}{K + 1} - g(s_t). \tag{14}$$

32

33 For a Markov sequential decision problem, the probability of the optimal path $L^* = \{s_0, \ldots, s_{n_L}\}$
34 being found by MCTS is:

$$P_g = P(s_0 \to s_1, s_1 \to s_2, \cdots, \to s_{n_L})$$

$$= \prod_{t=0}^{n_L - 1} P(s_t \to s_{t+1}), \tag{15}$$

35 where $P(s_t \to s_{t+1})$ denotes the probability that $s_{t+1}$ is selected while searching with state $s_t$ as the
36 root node. As shown in the right of Figure 2, assume state $s_t$ has $m$ children and the first child $s_{t+1}^1$
37 is in the optimal path. According to Assumption 4.2, $f$ values of $\{s_{t+1}^i, i = 2, 3, \cdots, m\}$ as well as
38 their descendant states are variables sampled from i.i.d. distribution $Pr(\mu_1^f, \sigma_1^2)$, because the optimal
39 path is in the subtree of $s_{t+1}^1$. Based on Lemma 2.3 and Eq. (14), $r_{t+1}^i + v(s_{t+1}^i) \sim \mathcal{N}(\mu_1^f - g(s_t),$
40 $\sigma_1^2/K_{t+1}^i)$ where $K_{t+1}^i$ is the simulation times of $s_{t+1}^i$. The $s_{t+1}^1$ is assumed to be the optimal child
41 of $s_t$ and $T(s_{t+1}^1)$ is composed by both the states in the optimal path and the ones not in the optimal
42 path. Assume there are $K_{t+1}^*$ descendants in the optimal path and $K_{t+1}^1 - K_{t+1}^*$ descendants are

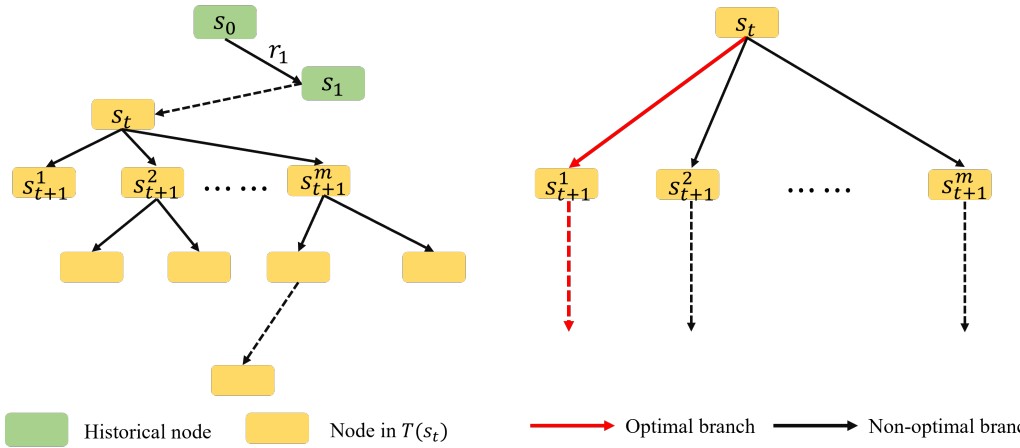

Figure 2: Left: a simulation of MCTS. $s_t$ is current state and $T(s_t)$ is the scouted subtree rooted with $s_t$, containing $K$ nodes. $s_{t+k}$ is the expanded node in $k$th simulation; Right: $s_{t+1}^1$ is selected after MCTS simulation rooted with $s_t$.

not in the optimal path. Let $T^*$ and $\overline{T^*}$ represent the descendant set in or not in the optimal path separately. $T(s) = T^* \cup \overline{T^*}$ and $T^* \cap \overline{T^*} = \emptyset$. In this situation:

$$
\begin{aligned}
\bar{f} &= \frac{\sum_{s' \in T(s_{t+1}^i)} f(s')}{K_{t+1}^1} = \frac{\sum_{s' \in T^*} f(s') + \sum_{s' \in \overline{T^*}} f(s')}{K_{t+1}^1} \\
&= \frac{K_{t+1}^*}{K_{t+1}^1} \mu_0^f + \frac{K_{t+1}^1 - K_{t+1}^*}{K_{t+1}^1} \frac{1}{K_{t+1}^1 - K_{t+1}^*} \sum_{s' \in \overline{T^*}} f(s').
\end{aligned}
\tag{16}
$$

$\sum_{s' \in \overline{T^*}} f(s')/(K_{t+1}^1 - K_{t+1}^*)$ is the mean of $K_{t+1}^1 - K_{t+1}^*$ variables sampled from $Pr(\mu_1^f, \sigma_1^2)$. Therefore, $\sum_{s' \in \overline{Tr^*}} f(s')/(K_{t+1}^1 - K_{t+1}^*) \sim \mathcal{N}(\mu_1^f, \sigma_1^2/(K_{t+1}^1 - K_{t+1}^*))$ according to the central limit theory in Lemma 2.3. Therefore,

$$
\frac{K_{t+1}^1 - K_{t+1}^*}{K_{t+1}^1} \frac{1}{K_{t+1}^1 - K_{t+1}^*} \sum_{s' \in \overline{T^*}} f(s') \sim \mathcal{N}(\frac{K_{t+1}^1 - K_{t+1}^*}{K_{t+1}^1} \mu_1^f, \frac{K_{t+1}^1 - K_{t+1}^*}{(K_{t+1}^1)^2} \sigma_1^2)
$$

$$
\begin{aligned}
\bar{f} &\sim \mathcal{N}(\frac{K_{t+1}^*}{K_{t+1}^1} \mu_0^f + \frac{K_{t+1}^1 - K_{t+1}^*}{K_{t+1}^1} \mu_1^f, \frac{K_{t+1}^1 - K_{t+1}^*}{(K_{t+1}^1)^2} \sigma_1^2) \\
&= \mathcal{N}(\frac{K_{t+1}^*}{K_{t+1}^1}(\mu_0^f - \mu_1^f) + \mu_1^f, \frac{K_{t+1}^1 - K_{t+1}^*}{(K_{t+1}^1)^2} \sigma_1^2)
\end{aligned}
$$

Therefore, $r_{t+1}^1 + v(s_{t+1}^1) = \bar{f} - g(s_t) \sim \mathcal{N}(\mu_1^f - g(s_t) + \frac{K_{t+1}^*}{K_{t+1}^1}(\mu_0^f - \mu_1^f), \frac{K_{t+1}^1 - K_{t+1}^*}{(K_{t+1}^1)^2} \sigma_1^2)$. When the simulation is finished, the decision is made based on

$$
a_{t+1} = \arg \max_i \left\{ r_{t+1}^i + v(s_{t+1}^i) \right\}.
\tag{17}
$$

In summary, for the optimal child $s_{t+1}^1$, we have $r_{t+1}^1 + v(s_{t+1}^1) \sim \mathcal{N}(\mu_1^f - g(s_t) + \frac{K_{t+1}^*}{K_{t+1}^1}(\mu_0^f - \mu_1^f), \frac{K_{t+1}^1 - K_{t+1}^*}{(K_{t+1}^1)^2} \sigma_1^2)$ and for state $s_{t+1}^i (i > 1)$, $r_{t+1}^i + v(s_{t+1}^i) \sim \mathcal{N}(\mu_1^f - g(s_t), \frac{\sigma_1^2}{K_{t+1}^i})$. The probability that the optimal child $s_{t+1}^1$ is selected is

$$
P(s_t \rightarrow s_{t+1}^1) = P(r_{t+1}^1 + v(s_{t+1}^1) = max_i \{r_{t+1}^i + v(s_{t+1}^i) | i \in [1, m]\})
\tag{18}
$$

According to the Lemma 2.2, we have

$$
P(s_t \rightarrow s_{t+1}^1) = \prod_{i=2}^m \left\{ 1 - \frac{1}{2} \exp \left\{ -\frac{\left( \frac{K_{t+1}^*}{K_{t+1}^1}(\mu_0^f - \mu_1^f) \right)^2}{2 \left( \frac{\sigma_1^2}{K_{t+1}^i} + \frac{(K_{t+1}^1 - K_{t+1}^*) \sigma_1^2}{(K_{t+1}^1)^2} \right) \cos^2 \xi_i} \right\} \right\}.
\tag{19}
$$

Let $K'$ denotes the least simulation times, that is $K_{t+1}^i \geq K'$ for all states. $\cos^2 \xi_i \leq 1$ always established. Therefore,

$$P(s_t \to s_{t+1}^1) \geq \left\{ 1 - \frac{1}{2} \exp \left\{ - \frac{\left( \frac{K_{t+1}^*}{K_{t+1}^1} (\mu_0^f - \mu_1^f) \right)^2}{2 \left( \frac{1}{K'} + \frac{K_{t+1}^1 - K_{t+1}^*}{(K')^2} \right) \sigma_1^2} \right\} \right\}^{m-1}. \tag{20}$$

Based on Eq 15, the probability of the optimal path $L^* = \{s_0, \cdots, s_{n_L}\}$ being found by MCTS is

$$Pg = \prod_{t=0}^{n_L-1} P(s_t \to s_{t+1}) \geq \prod_{t=0}^{n_L-1} \left\{ 1 - \frac{1}{2} \exp \left\{ - \frac{\left( \frac{K_{t+1}^*}{K_{t+1}^1} (\mu_0^f - \mu_1^f) \right)^2}{2 \left( \frac{1}{K'} + \frac{K_{t+1}^1 - K_{t+1}^*}{(K')^2} \right) \sigma_1^2} \right\} \right\}^{m-1}$$

$$= \prod_{t=1}^{n_L} \left\{ 1 - \frac{1}{2} \exp \left\{ - \frac{[b_t(\mu_0^f - \mu_1^f)]^2}{2(1/K' + m_t)\sigma_1^2} \right\} \right\} \tag{21}$$

where $b_t = K_t^*/K_t^1$, $m_t = (K_t^1 - K_t^*)/(K_t^1)^2$, and $K'$ denotes the least simulation times, that is $K_t^i \geq K'$ for an arbitrary state. If MCTS's simulation times $K$ is large enough, every child will be visited enough times because of the exploration term in Eq. (6), that are $K' \to +\infty$, $m_t \to 0$, and $b_t$ approaches a constant when $K \to +\infty$. If $b_t = \infty$, the optimal branch will always be selected according to Eq. (6) until $b_t$ becoming a limited constant. Therefore, we have

$$\lim_{K \to \infty} P_g \geq \prod_{t=1}^{n_L} \left\{ 1 - \frac{1}{2} \exp \left\{ - \frac{[b_t(\mu_0^f - \mu_1^f)]^2}{2(1/K')\sigma_1^2} \right\} \right\} \tag{22}$$

Theorem 4.3 has been proven.

Table 1: Winning rate of PCZero against AlphaZero without PC (in percentage %).

| BoardSize | $8 \times 8$ | | $9 \times 9$ | | $13 \times 13$ | |
|---|---|---|---|---|---|---|
| Player | Greedy Player | MCTS Player | Greedy Player | MCTS Player | Greedy Player | MCTS Player |
| $\lambda = 0.1$ | **53.1** | **56.3** | 51.9 | 56.8 | 47.6 | 49.4 |
| $\lambda = 0.5$ | 49.2 | 54.7 | **54.3** | 49.4 | 51.5 | 49.1 |
| $\lambda = 1.0$ | 48.4 | 50.0 | 53.1 | 54.3 | 44.7 | 53.6 |
| $\lambda = 2.0$ | 51.6 | 53.1 | 53.1 | **59.9** | **52.1** | **63.9** |

Table 2: Test results with 32 seeds, presented as mean±standard deviation.

| Game | EfficientZero[†] | GW-PCZero | Game | EfficientZero[†] | GW-PCZero |
|---|---|---|---|---|---|
| Alien | $850.6 \pm 339.2$ | $699.7 \pm 130.7$ | Amidar | $60.6 \pm 2.42$ | $97.0 \pm 12.3$ |
| Assault | $994.8 \pm 181.4$ | $1224.1 \pm 371.2$ | Asterix | $17734.4 \pm 2921.9$ | $14771.9 \pm 5018.8$ |
| BankHeist | $276.9 \pm 40.4$ | $207.2 \pm 59.8$ | BattleZone | $15875.0 \pm 4614.9$ | $13500.0 \pm 6557.4$ |
| Boxing | $28.2 \pm 7.2$ | $41.6 \pm 11.7$ | Breakout | $366.7 \pm 56.1$ | $450.0 \pm 160.8$ |
| ChopperCmd | $818.8 \pm 323.5$ | $1150.0 \pm 362.3$ | CrazyClimber | $8059.4 \pm 2242.9$ | $9734.4 \pm 4233.3$ |
| DemonAttack | $7940.8 \pm 3835.9$ | $24074.1 \pm 15593.6$ | Freeway | $0.0 \pm 0.0$ | $0.0 \pm 0.0$ |
| Frostbite | $229.1 \pm 19.9$ | $249.7 \pm 16.3$ | Gopher | $1325.6 \pm 638.3$ | $1286.9 \pm 803.1$ |
| Hero | $7537.2 \pm 81.7$ | $8171.3 \pm 795.3$ | Jamesbond | $300.0 \pm 179.0$ | $525.0 \pm 252.5$ |
| Kangaroo | $525.0 \pm 277.3$ | $262.5 \pm 145.2$ | Krull | $3818.5 \pm 600.5$ | $7782.0 \pm 1018.6$ |
| KungFuMaster | $8956.3 \pm 1816.4$ | $20543.8 \pm 5216.1$ | MsPacman | $967.5 \pm 320.9$ | $1594.1 \pm 746.8$ |
| Pong | $15.6 \pm 4.5$ | $19.8 \pm 1.2$ | PrivateEye | $0.0 \pm 0.0$ | $96.9 \pm 17.4$ |
| Qbert | $8120.3 \pm 632.2$ | $13651.6 \pm 2216.1$ | RoadRunner | $3443.7 \pm 1058.6$ | $16809.4 \pm 3635.1$ |
| Seaquest | $478.1 \pm 82.8$ | $768.1 \pm 210.8$ | UpNDown | $7592.5 \pm 3997.6$ | $12344.7 \pm 5173.7$ |

# 3 Investigation of $\lambda$ on board games

Table 1 shows the winning rate of PCZero against AlphaZero without path consistency in Hex game with different board sizes. The larger the size of the board, the more complex the problem becomes. We can see that the game with a smaller board size should have a smaller PC loss weight $\lambda$ and the game with a larger board size should have a larger $\lambda$ to fully utilize path consistency.

## 4 Variance of the result

Tested with 32 seeds, result with standard deviation is summarized in Table 2.

## 5 Hyper-parameters setting

Neural network in this paper is the same as EfficientZero. Hyper-parameters are listed in Table 3, which are the same with EfficientZero except that training steps are changed from $120k$ to $60k$ and the off-policy value correction is disabled. State value is reanalyzed with value network instead of MCTS's root value.

Table 3: Hyper-parameters of the learning process

| Parameter | Setting |
|---|---|
| Observation down-sampling shape | $96 \times 96$ |
| Frames stacked | 4 |
| Frames skip | 4 |
| Discount factor | $0.997^4$ |
| Batch size | 256 |
| Optimizer | SGD |
| Learning rate | $0.2 \rightarrow 0.02$ |
| Momentum | 0.9 |
| Weight decay | 0.0001 |
| Max gradient norm | 5 |
| Priority exponent | 0.6 |
| Priority correction | $0.4 \rightarrow 1$ |
| Training steps | $60k$ |
| Evaluation episodes | 32 |
| Min replay size for sampling | $2,000$ |
| Self-play network updating interval | 100 |
| Target network updating interval | 200 |
| Unroll steps | 5 |
| TD steps | 5 |
| Policy loss coefficient | 1.0 |
| Value loss coefficient | 0.25 |
| Self-supervised consistency loss coefficient | 2.0 |
| Value prefix loss coefficient | 1.0 |
| Dirichlet noise ratio | 0.3 |
| Number of simulations in MCTS | 50 |
| Reanalyzed policy ratio | 0.99 |
| Selfplay max moves | $108,000$ |
| Test max moves | $12,000$ |
| LSTM horizon | 5 |
| LSTM hidden size | 512 |
| Network parameter initialize zero | True |
| Clip reward | True |
| RGB image based | True |
| Do self-supervised consistency | True |
| Use value-prefix | True |
| MCTS Off-policy value correction | False |

## 6 Comparison of evaluation curves

Learning curves of all 26 Atari games are displayed in Figure 3, 4 & 5.

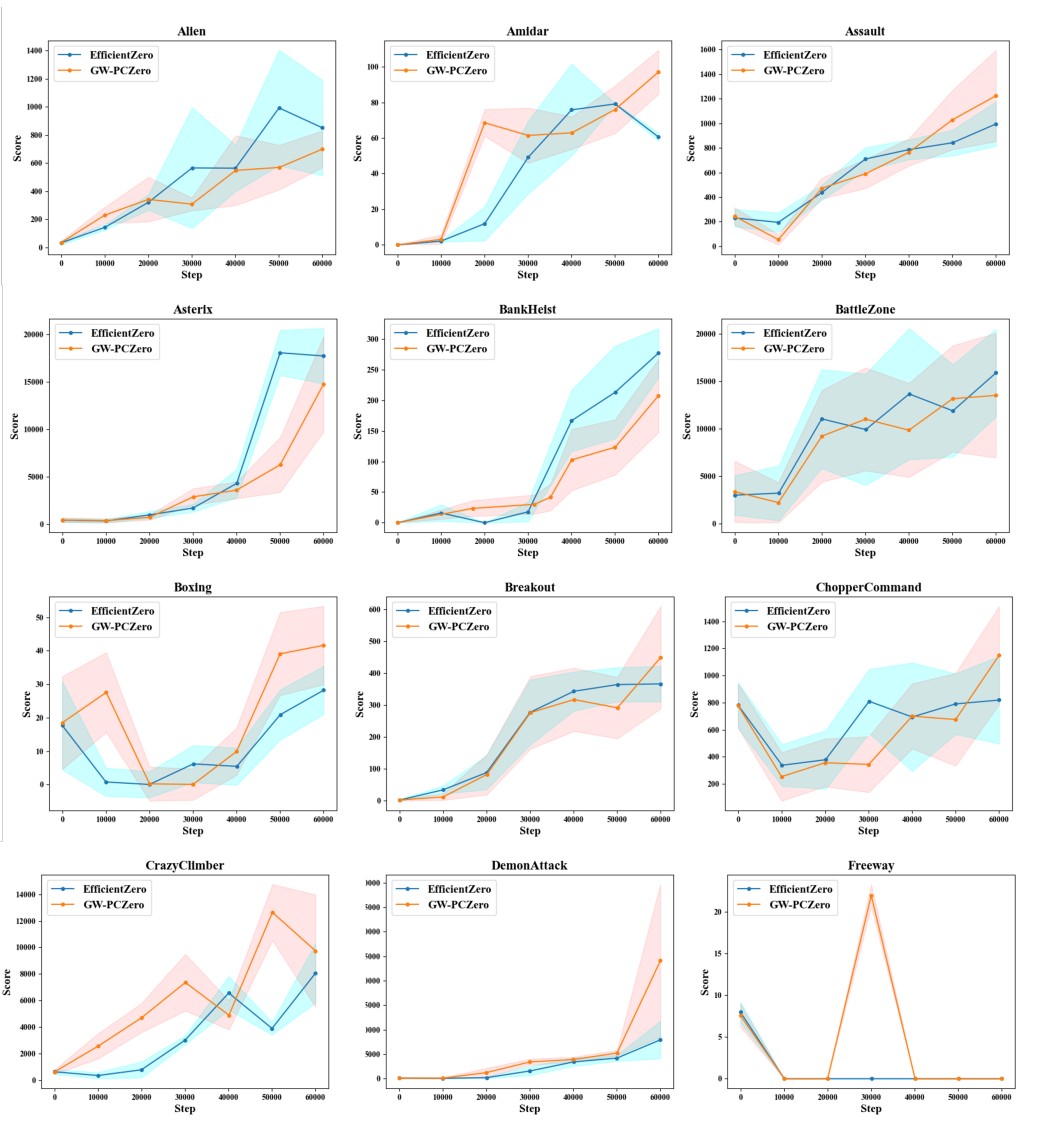

Figure 3: Learning curves (Part 1)

# 7 Experiment on more games

## 7.1 Hex game

In this section, the idea of weighting path consistency is applied to PCZero on $13 \times 13$ Hex game. In PCZero [6], the learning target is calculated as the mean of $l$ upstream states and $k$ downstream states in Eq (23).

$$t_{PC}(s_t) = \frac{1}{l+k} \sum_{i=-l}^{k} v(s_{t+i}).$$ (23)

Considering weighting mechanism, the learning target is calculated by:

$$t_{PC}(s_t) = \sum_{i=-l}^{k} w_i v(s_{t+i}) / \sum_{i=-l}^{k} w_i,$$ (24)

where $w_i$ is linear decay weight. As the distance from $s_t$ increases, $w_i$ decreases proportionally as shown in Eq (25)

$$w_i = b_0 - a_0 \times |i|.$$ (25)

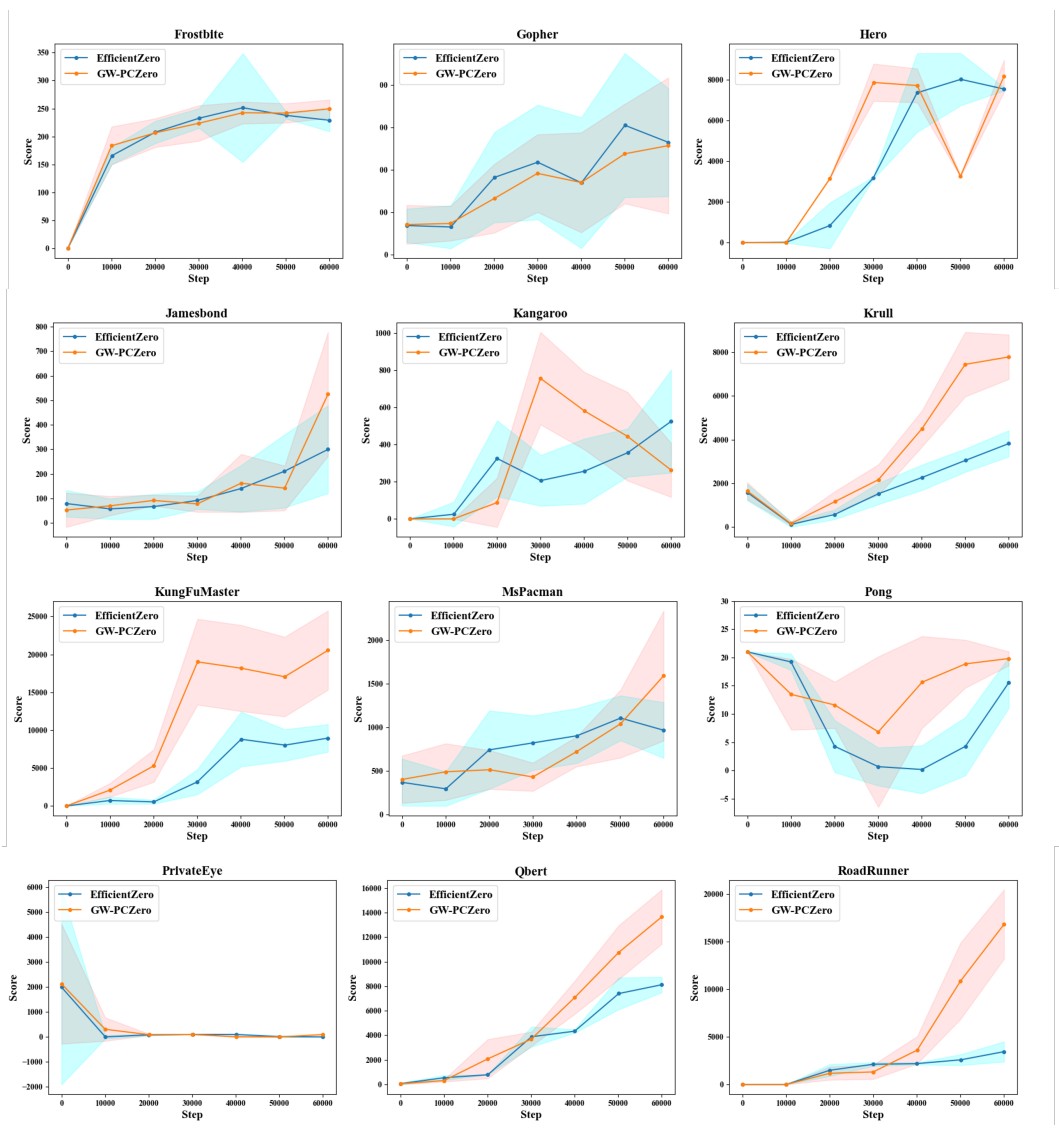

Figure 4: Learning curves (Part 2)

In the experiment, $b_0 = 1.0$ and $a_0 = 0.1$. Trained with the same dataset with PCZero, which is consist of $900k$ selfplay games, Weighted PCZero beats the original PCZero with $175 : 163$ score, when the simulation times of MCTS is $800$, demonstrating that weighting mechanism is also beneficial to PCZero and it deserves further investigation.

## 7.2 Classic control problem

We also investigate the idea of generalized weighted path consistency on MuZero [3]. The implementation of PC is exactly the same as GW-PCZero, except that the underlying EfficientZero has been replaced with MuZero, which is available in `https://github.com/koulanurag/muzero-pytorch`. The CartPole problem is used for comparison, for which the goal is to balance the pole by applying forces in the left and right direction on the cart. The learning cures are displayed in Figure 6. On the left is MuZero without reanalyzing. on the right is MuZero with reanalyzing and the proportion of reanalyzing is $0.99$. Path consistency significantly improves the model's performance in both cases. The idea of generalized weighted path consistency is also effective for MuZero.

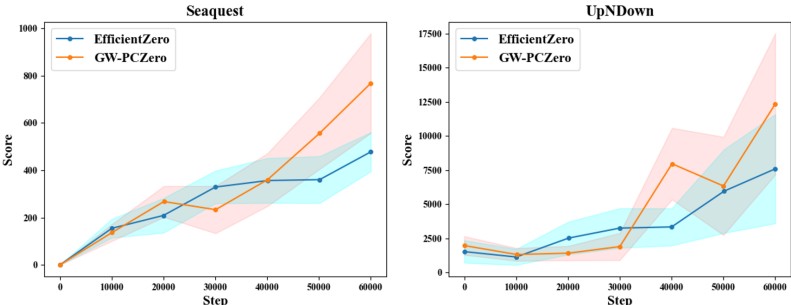

Figure 5: Learning curves (Part 3)

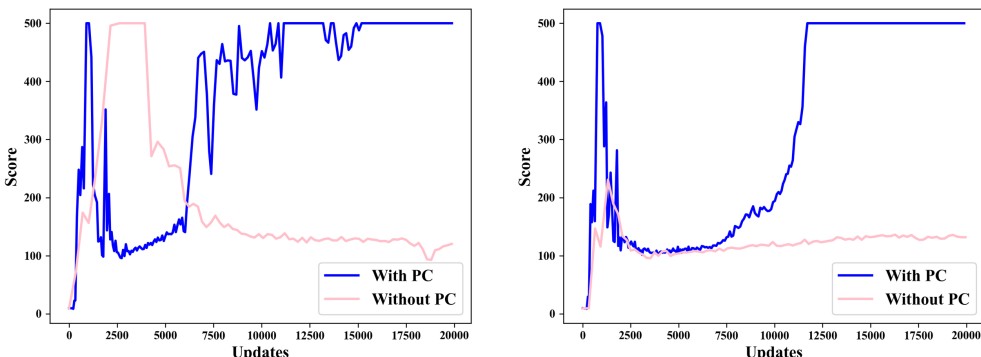

Figure 6: Learning curves for MuZero with and without path consistency. (Left: MuZero without reanalyze; Right: MuZero with $0.99$ reanalyze rate.)

## 8    Pseudocode for GW-PCZero

In this section, we will provide a brief summary of the pseudocode for GW-PCZero algorithm. As shown in Algorithm 1, the entire training process can be divided into three parts. The first part involves collecting game frames by employing a MCTS player guided by the policy and value network. The second part entails reanalyzing the collected states in the playing path to generate labels for training the model. This process is illustrated in Algorithm 1, and the PC target is prepared by calculating the weighted average of the $f$ values along the path, as depicted in Algorithm 3. The third part entails updating the policy model and value model using the prepared data, where the loss function is defined in Equation (2). In this equation, $\mathcal{L}_{RL}$ is the same as that used in EfficientZero [5], and $\mathcal{L}_{PC}$ is defined in Equation (10).

---

**Algorithm 1:** Framework for GW-PCZero

---

**Input**: Training steps $N$
**Output**: Policy and value network.
 1: Initialize policy network $\pi$ and value network $v$.
 2: $n \leftarrow 0$
 3: **while** $n < N$ **do**
 4:     Collect playing game frames with MCTS player guided by $\pi$ and $v$.
 5:     Prepare learning target by reanalyzation with MCTS in Algorithm 2.
 6:     Update $\pi$ and $v$ using the loss function defined in Eq. (2).
 7: **end while**
 8: **return** $t_{PC} = T / \sum w_i$.

---

---
**Algorithm 2:** Sample Preparation for GW-PCZero
---
**Input**: Replay buffer $\mathcal{R}$, Unrolled steps $l$.
**Output**: $(\pi, z, t_{PC})$.
 1: Sample unrolled sequences with $l + 1$ states from $\mathcal{R}$.
 2: **for** each sampled sequence **do**
 3:     Reanalyze policy target $\pi$ by MCTS.
 4:     Recalculate value target $z$ by bootstrapping in Eq. (9).
 5:     Estimate PC target $t_{PC}$ according to Algorithm 3.
 6: **end for**
 7: **return** Tuple $(\pi, z, t_{PC})$.
---

---
**Algorithm 3:** Weighted PC target $t_{PC}$ estimation
---
**Input**: $S = \{s_t, r_{t+1}, s_{t+1}, \cdots, s_{t+l}\}$, value function $v(s)$ and weights $w = \{w_0, w_1, \cdots, w_l\}$.
**Output**: Target $t_{PC}$.
 1: Initialize $T = 0$.
 2: **for** each state $s_{t+i}$ in S **do**
 3:     $T = T + w_i \times \left[ \sum\limits_{j=1}^{i} r_{t+j} + v(s_{t+i}) \right]$
 4: **end for**
 5: **return** $t_{PC} = T / \sum w_i$.
---