# OpenReview forum: "Generalized Weighted Path Consistency for Mastering Atari Games"
_NeurIPS.cc/2023/Conference — NeurIPS 2023 poster_

### Official Review · Reviewer_Dd1j · 2023-06-27

**Soundness:** 3 good
**Presentation:** 3 good
**Contribution:** 3 good
**Rating:** 7
**Confidence:** 4

**Summary:**

This paper proposed Generalized Weighted PCZero (GW-PCZero), which builds on EfficientZero and PCZero. The goal is to generalize the implementation of PCZero from board games to Atari games, which is achieved by adding the theorem “path consistency” to EfficientZero, extending the previous idea from PCZero. This paper further applies a weighting mechanism to path consistency, which calculates more accurate targets for agents to learn. Furthermore, this paper proves that neural-guided MCTS is guaranteed to find the optimal solution under PC constraint, providing a theoretical foundation for PC. The experiments show that under Atari 100k benchmark, GW-PCZero achieves 198% mean human normalized performance, slightly higher than the SOTA EfficientZero (194%). More importantly, GW-PCZero only consumes 25% computational resources of EfficientZero.

**Strengths:**

1. GW-PCZero achieves slightly higher performance under Atari 100k benchmark than the SOTA EfficientZero while consuming only 25% computational resources of EfficientZero.
2. GW-PCZero generalizes the implementation of path consistency from board game to the case where the environment emits immediate reward, such as Atari game.
3. This paper proves that neural-guided MCTS is guaranteed to find the optimal solution under PC constraint.
4. The paper provides rich experimental data. In addition to the Atari 100k benchmark, the authors conducted experimental analyses on board games (Hex) by combining PCZero with the weighting mechanism. Furthermore, experiments were conducted on the Classic control problem - Cartpole. The diverse range of experimental environments shows the generality of this method.


**Weaknesses:**

It appears that the proposed method is primarily a combination of two previous works, PCZero and EfficientZero, without introducing significantly novel ideas or concepts in the algorithm itself. However, overall, I would still think it is valuable to obtain the state-of-the-art with the combination.


**Questions:**

1. In Appendix 7.1, regarding the board game Hex, you mentioned that "Weighted PCZero beats the original PCZero with a score of 175:163." However, based on the score of 175:163, it seems that the two are relatively close in performance, and it is not evident from this result alone the advantage of the Weighted mechanism. Probably, you want to show some confidence error.
2. In Appendix 7.2, the experiments for the Classic control problem only include Cartpole, which is a relatively simple environment. Are there any other experiments that demonstrate the applicability of GW-PCZero to more complex control problems? For example, EfficientZero applied their work to Deepmind Control 100k, and it would be beneficial if you could apply your method to Deepmind Control 100k and compare it with their results.


**Limitations:**

N.A.

---

> ### Author Rebuttal · Authors · 2023-08-09
>
> We thank the reviewer for constructive comments and suggestions, and we will carefully revise the paper accordingly. We provide a detailed response to each question below.
> Q1: In Appendix 7.1, regarding the board game Hex, you mentioned that "Weighted PCZero beats the original PCZero with a score of 175:163." However, based on the score of 175:163, it seems that the two are relatively close in performance, and it is not evident from this result alone the advantage of the weighting mechanism
> A1: The effectiveness of the weighting mechanism becomes more evident when the game trajectories are of low-quality. Here, we demonstrate the effect by training the model gradually on the self-generated games during the reinforcement learning process. In the early stages of training, the model's playing ability is relatively weak, and the quality of the generated game playing trajectories is poor. When trained on the 50k games, GW-PCZero beats PCZero with a score of 199:139. When trained on the first 250k games, GW-PCZero beats PCZero with a score of 216:122. It is noted that the performance gap between these two models becomes more evident, i.e., the advantage of the weighting mechanism becomes more obvious.
> Q2: In Appendix 7.2, the experiments for the Classic control problem only include Cartpole, which is a relatively simple environment. Are there any other experiments that demonstrate the applicability of GW-PCZero to more complex control problems?
> A2: Thank you for your valuable suggestion. We are working on applying our GW-PCZero to the DeepMind Control (DMC) 100k tasks. Currently, our GW-PCZero is built on EfficientZero, but the source code of EfficientZero for the DMC task is not available from the original authors. We have not yet reproduced the results of EfficientZero on the DMC tasks. In the future, we will make every effort to successfully perform our GW-PCZero on the DMC tasks or find another way to demonstrate that the idea of Path Consistency (PC) also works on the DMC tasks.

---

### Official Review · Reviewer_hxZc · 2023-06-28

**Soundness:** 3 good
**Presentation:** 3 good
**Contribution:** 3 good
**Rating:** 7
**Confidence:** 3

**Summary:**

This paper proposes GW-PCZero,  an RL algorithm based on neural-guided Monte Carlo Tree Search (MCTS). GW-PCZero adopts the idea of Path Consistency (PC) from prior work, i.e., a regularizer that encourages evaluation function to be consistent throughout an optimal path, to improve sample efficiency. Beyond this, GW-PCZero generalizes to the environment with reward given before the end of the episode, and appends a weight in the regularizer that decreases with the number of steps of the episode to account for increasing uncertainty later in the episode. The paper proves that the probability of finding the optimal path is lower-bounded, and achieves comparable / marginally better performance than the state-of-the-art (EfficientZero) with much less computational cost on many Atari environments.

**Strengths:**

**1. The writing is easy to follow and the idea is clearly conveyed.** There are many designs of the prior work that needs to be introduced, such as neural MCTS, path consistency, and re-analyzation of both MuZero and EfficientZero; moreoever, the motivations of PCZero and EfficientZero have to be clearly stated for the readers to understand the line of work. This paper does a good job in the first three section where the concepts are well-explained to the readers. The theorems and assumptions are also clearly stated in the methodology section.

**2. The experiment results are solid and convincing.** The superior performance of GW-PCZero has been evaluated on Hex and many Atari environments, and the ablation on the most important hyperparameters, which are the PC weight factor and $\lambda$, are provided in detail. Furthermore, the authors have submitted the code, which makes the result more convincing.


**3. The algorithm has theoretically proved performance, which is a valuable contribution.** The paper has proved that GW-PCZero has a bound for probability of finding the optimal path, which is the first theoretical result in this line of work. I am convinced that the proving technique utilized in the paper would be beneficial for further theoretical research into neural MCTS.

**Weaknesses:**

**1. It seems that the result is somewhat sensitive to the choice of weighting hyperparameters.** In table 3, the scores are quite different with adjacent choice of $c_a$, and different environment seems to have very different optimal $c_a$. This is also the case in the selection of $\lambda$ in table 1 of the appendix, and there is no monotonicity exhibited on either side of the optimal hyperparameter (e.g. 8*8 and 9*9 MCTS player).

**2. Other minor problems:**

a) The second part of Table 3 in the main paper should also have a horizontal line between the first row and the rest of the table.

b) Should Eq. 16 be clipped with 0 from below? by line 281 we know that when $i>10$ the weight will be negative, but it does not make sense to discourage path consistency at any time.

**Questions:**

I have four questions for this paper:

1. The experiment results only show the effect of How sensitive the $\lambda$ is on the board games, which is a special case. How sensitive would $\lambda$ on the Atari environment be?

2. In section 4.4, the $\lambda$ is chosen according to the ratio between prior work performance and expert (human) performance. However,  the evaluation of expert (human) performance might not be available in real-life applications. Have the author considered to use a adaptable $\lambda$ that dynamically changes during the training process? (I noticed that the authors mention "systematic investigation on automatic weighting methods in the future", but this is about a different hyperparameter.)

3. The author claims that they can do marginally better than EfficientZero with a quarter of MCTS runs. While this is exciting, a constant factor improvement in the time complexity might be cancelled out by implementation and more values to calculate (e.g. the extra regularizer term). Is it possible for the authors to provide actual wall clock time for each method, or to explain that your method do not have a significant overhead compared to EfficientZero for each MCTS run?

4. It is a little surprising that in the Appendix section 7.2, MuZero without Path Consistency (PC) cannot deal with environments as simple as cartpole, either with or without reanalyzation. Also, there are two interesting phenomenon about this figure, which are 1) for all curves, the rewards first increase quickly, then decrease to a relatively low level, and 2) with no reanalization, MuZERO without PC even seems to work better. Could the authors explain the figure more carefully?

There are also three suggestions besides addressing the questions above and in the weakness section:

1. Limitation section are missing in the paper. While there are some places that implicitly mentions the limitation (e.g. line 327, "it deserves a systematic investigation on better automatic weighting ... in the future"), I suggest the author to think of more possible limitations (see limitation section) and summarize them in a separate paragraph.

2. I suggest the authors to append a pseudocode in the appendix to more clearly shows how GW-PCZero works; it is even better if the authors could highlight the difference in the pseudocode with prior work such as PCZero.

3. The authors claim that there is no immediate ethical or social impact of this work. While this is true, there are still broader impacts of the paper that still needs to be considered, such as potential misuse of the automated technology and potential job loss that needs to be considered.

**Limitations:**

The authors implicitly mentioned limitations (line 327, "it deserves a systematic investigation on better automatic weighting ... in the future"). However, I suggest the authors to think more about limitations, such as hyperparameter sensitivity, manual selection of hyperparameters, possible limitation on future application by the nature of neural-guided MCTS, etc., and summarize them into a separate paragraph. As for potential negative societal impact, the author claims that there are no immediate negative societal impact. While this is true, I encourage the authors to be aware of the broader impact of their work on automated decision making.

---

> ### Author Rebuttal · Authors · 2023-08-09
>
> Thank you so much for your valuable suggestions. We will incorporate more discussions about algorithm’s limitations in the paper. For example, similar to many existing algorithms, the performance of the current implementation of GW-PCZero relies on the selection of hyperparameters. In the long term, efficient reinforcement learning algorithms may be applied in many real-world scenarios, such as factory robots, which may lead to worker displacement. The complete pseudocode will be provided in the appendix. In the following, we provide a detailed explanation of the raised concerns.
> Q1: The experiment results only show the effect of How sensitive the λ is on the board games, which is a special case. How sensitive would $\lambda$ on the Atari environment be?
> A1: The conclusion drawsn from board games also hold on Atari games. We add an experiment of  fixing $\lambda$ at 0.2 or 0.35, and the results are reported below. We observe that out of the 11 simple games, 9 achieved better results with smaller $\lambda$. This is consistent to the practical experience given in this paper, i.e., it is better to set a small $\lambda$ for the relatively simple games whose the action space is less than 18.
> |Game	|	$\lambda$=0.2	|	$\lambda$=0.35	|	Game	|	$\lambda$=0.2	|	$\lambda$=0.35	|
> | :-:   |   :-:             | :-:               |:-:   |   :-:             | :-:               |
> |Breakout	|	450.0		|384.1|Pong		|19.8			|6.7|
> |Qbert		|13651.6		|7439.1	|		Assault	|	1224.1		|1350.2|
> |UpNDown	|12344.7		|3932.5	|		Asterix	|	14771.9		|9000.0|
> |CrazyClimber|	9734.4		|6665.6	|		DemonAttack|	24074.1		|12116.6|
> |MsPacman	|1594.1		|805.0|			Amidar		|97.0	|	160.1|
> |KungFuMaster|	20543.8	|10400.0|
>
> Q2: In section 4.4, the $\lambda$ is chosen according to the ratio between prior work performance and expert (human) performance. However, the evaluation of expert (human) performance might not be available in real-life applications. Has the author considered to use an adaptable $\lambda$ that dynamically changes during the training process?
> A2: Thanks for your valuable suggestions. Making the coefficient $\lambda$ dynamically adapted during the training process is under our consideration. We have been trying to reduce $\lambda$ gradually as the training proceeds. In the current version of the paper, we suggest to use a small $\lambda$ for the relatively simple games and a large $\lambda$ for the complex games according to our practical experience. We agree that it is challenging to assess the difficulty of each Atari game,  and the ratio given in the paper is an approximate measure. In practical applications, other factors can also be adopted to evaluate the difficulty of the game, such as its state-space complexity, game-tree complexity, and so on. It deserves further investigations in the future.
> Q3: The author claims that they can do marginally better than EfficientZero with a quarter of MCTS runs. While this is exciting, a constant factor improvement in the time complexity might be cancelled out by implementation and more values to calculate (e.g. the extra regularize term). Is it possible for the authors to provide actual wall clock time for each method, or to explain that your method do not have a significant overhead compared to EfficientZero for each MCTS run?
> A3: The regularization term induced by Path Consistency (PC) for the loss function increases some extra computation in the training phase but not much. We report the wall clock times by taking the game of Breakout as an example due to the time limitation in the rebuttal period. GW-PCZero with 60k training steps spends 254 minutes on the training process. If we remove the PC term, the training time of GW-PCZero reduces to 230 minutes. That is, the extra training computation due to the PC term costs 24 minutes, which is a 10.4% (24/230) increment in the wall clock time. Moreover, since the PC term does not affect the MCTS runs, the computational cost of GW-PCZero for each MCTS run is the same as that of EfficientZero. Therefore, the extra computational time due to the PC term is relatively a very small proportion of the overall cost. We will report the actual wall clock time for all the methods on all Atari games in the revised version of the paper.
> Q4: It is a little surprising that in the Appendix section 7.2, MuZero without Path Consistency (PC) cannot deal with environments as simple as cartpole, either with or without reanalyzation. Also, there are two interesting phenomenon about this figure, which are 1) for all curves, the rewards first increase quickly, then decrease to a relatively low level, and 2) with no reanalyzation, MuZERO without PC even seems to work better. Could the authors explain the figure more carefully?
> A4: As discussed in the paper, PC can enhance learning efficiency. Therefore, as shown by the learning curves in the Appendix Section 7.2, MuZero with PC learns to increase the score or reward faster than the version without PC in the early steps. Although the score curve of the PC version drops down later, it climbs back and stays at the top robustly, whereas the non-PC version drops down and stays at a low score level. What’s more, reanalyzation is one of the sources of uncertainty, and it is more helpful for the weighting mechanism in PC to take effect. The first phenomenon of the fluctuation in the score curves may be attributed to the implementation of MuZero. For the second phenomenon, the PC version of MuZero achieves a much higher score than the non-PC version as learning proceeds to be stable.
> Q5: Should Eq. 16 be clipped with 0 from below?
> A5: Thank you for pointing out this issue. We will revise it accordingly.

---

> > ### Comment · Reviewer_hxZc · 2023-08-11
> > **Response to Rebuttal**
> >
> > Thanks for your detailed repsonse; I think they addresses my concern well, though it is a pity that the proposed algortihm seems to be somewhat sensitive with respect to $\lambda$ on the Atari environment. I have one follow up question: Is the wall clock time of EfficientZero available?

---

> > > ### Author Response · Authors · 2023-08-12
> > >
> > > Thank you very much for your response. Taking the game of Breakout as an example, EfficientZero needs 230 minutes if the updating steps is 60k and the MCTS root value correction is disabled. If MCTS root value correction is allowed and the updating steps is still 60k, the spent time of EfficientZero will be increased to 275 minutes. If the updating step is 120k and  the MCTS root value correction is disabled, EfficientZero will need 503 minutes. Therefore, the extra training computation due to the MCTS root value correction costs 45 minutes, which is a 19.6% (45/230) increment in the wall clock time. Because the calculations can be done in parallel, the time is not doubled. If the number of update steps is doubled, the required training time is roughly doubled.

---

> > > > ### Comment · Reviewer_hxZc · 2023-08-12
> > > > **Further Response**
> > > >
> > > > Thanks a lot for the response; I appreciate the authors' effort in addressing my concerns and decide to keep my score of 7.

---

### Official Review · Reviewer_s6tP · 2023-07-05

**Soundness:** 2 fair
**Presentation:** 3 good
**Contribution:** 1 poor
**Rating:** 4
**Confidence:** 5

**Summary:**

This paper proposes a model-based RL method called GW-PCZero, which is built on EfficientZero and generalizes the path consistency (PC) constraint from board games with zero immediate rewards to environments with non-zero immediate rewards. The authors introduce a weighting average mechanism and use the mean f value of states along the path as the target for the PC constraint.

Although the authors did comprehensive experiments to demonstrate that GW-PCZero slightly outperforms EfficientZero on the Atari 100k benchmarks with 26 games with less computation, they missed a published paper SCZero whose idea is essentially the same as the proposed GW-PCZero. This makes the novelty and comprehensiveness of GW-PCZero should be questioned.

Meanwhile, since the author uses PC constraint on a off-policy RL algorithm EfficientZero, the off-policy issues may have significantly negative impacts on the effectiveness of PC loss. In the setting of off-policy RL, the author's claim on global constraint of PC loss should be questioned. The authors only make limited discussion and proposed a simple linear weighting trick, which causes PC loss to degenerate into the SC loss.

**Strengths:**

1. The paper extends the path-consistency to environments with non-zero immediate rewards.
2. The paper uses significantly less computational resources.

**Weaknesses:**

The most critical issues of this paper:
1. The proposed GW-PC loss is extremely similar to a published paper 'Self-Consistent Models and Values'(Neurips 2021). The PC loss can be converted to SC loss easily as
$l^{SC-residual}=\sum_{k=0}^K (\hat{r}(s_k, a_k)+\gamma \hat{v}(s_{k+1})-\hat{v}(s_k))=\sum_{k=0}^K (g(s_{k+1})+\hat{v}(s_{k+1})-g(s_k)-\hat{v}(s_k))=l^{PC}$
However, it seems that the author did not notice SCZero paper at all, no citing or taking SCZero as the most important baseline.
2. Setting path consistency on a off-policy algorithm like EfficientZero is not reasonable. For tasks with nonzero immediate rewards like Atari Games, the old state transitions collected by the old policy brought significant off-policy issue, which makes the proposed PC loss  hardly be a global constraint as the authors claimed. Considering the off-policy issue and computational limitations, authors proposed a sliding window and a linear weighting trick to reduce the impact of the marginal states within the sliding window. This makes the proposed PC loss has little improvement compared to the previous SC loss, no matter in terms of fundamental ideas or mathematics.

Minors:
1. The paper didn't report the median human normalized score.
2. The coefficient of PC loss is required to be adjusted according to the task. Generally, this is not an common hyperparameter in most RL settings. And the performance should not be sensitive to the coefficient of PC loss.
3. The paper didn't compare it with other data-efficient RL algorithms like IRIS and Dreamer.
4. The paper over-claims the contribution. GW-PCZero only slightly outperform the full-version EfficientZero(with 120k training steps).

**Questions:**

1. Can you report the median human normalized score?
2. Can you report the performance of GW-PCZero(c_\alpha)? and also the performance of Dreamer and IRIS if possible.
3. Can you report a performance with fixed coeff of PC loss? like \lambda=0.2 or 0.4
4. Can you report the performance of GW-PCZero with 120k training steps?
5. Can you provide some experiment results that shows PC loss better than SC loss within Atari 100K setting?

**Limitations:**

The paper's novelty should be questioned, since its PC loss is basically a 'reward-sum' version of DeepMind's SC loss published in 'Self-Consistent Models and Values', Neurips 2021.
The coefficient of PC loss needs to be adjusted according to the task. Generally, this is not an common hyperparameter in most RL settings. And the performance should not be sensitive to the coefficient of PC loss.

---

> ### Author Rebuttal · Authors · 2023-08-09
>
> We sincerely appreciate your valuable suggestions, and in the following we address the concerns you have raised.
> Q1: Can you report the median human normalized score?
> A1: The median human normalized scores are 0.399 and 0.388 for GW-PCZero and EfficientZero, respectively. The results are obtained under the same training conditions. The human normalized scores of GW-PCZero for each of the 26 games are given in decreasing order as follows. It is observed that GW-PCZero outperforms human on 11 games (with score > 1). The median is computed by the average between the $13^{th}$ (Gopher) and the $14^{th}$ (BattleZone).
> |Game	|	Score	|	Game	|	Score	|	Game	|	Score	|
> | :-:   |   :-:             | :-:               |:-:   |   :-:             | :-:               |
> |Breakout	|	15.57		|DemonAttack|	13.15		|Krull		|5.79|
> |Boxing		|3.46			|RoadRunner	|2.14			|Assault		|1.93|
> |Asterix	|	1.76		|	Jamesbond	|1.38		|	Pong		|1.15|
> |UpNDown	|1.06			|Qbert		|1.01			|KungFuMaster	|0.90|
> |Gopher		|0.48			|BattleZone	|0.32			|BankHeist	|0.26|
> |Hero		|	0.24		|	MsPacman	|0.19		|	Alien		|0.07|
> |Kangaroo	|	0.07		|	Amidar		|0.05		|	ChopperCmd	|0.05|
> |Frostbite	|	0.04		|	Seaquest		|0.02	|		PrivateEye|	0.00|
> |Freeway	|	0.00		|	CrazyClimber	|-0.04|
>
> Q2: Can you report the performance of GW-PCZero($c_\alpha$)? and also the performance of Dreamer and IRIS if possible.
> A2: The performance of GW-PCZero($c_\alpha$) is reported as follows. Experiment results indicates that proper weighting mechanism is beneficial to improve the performance, and we will conduct a more detailed and rigorous study of the weighting mechanism in the future.
> |Game	|	Score	|	Game	|	Score	|	Game	|	Score	|
> | :-:   |   :-:             | :-:               |:-:   |   :-:             | :-:               |
> |Alien	|	982.5		|Amidar		|97.0		|	Assault		|1224.1|
> |Asterix|		22131.3	|	BankHeist|	207.2	|	BattleZone	|16812.5|
> |Boxing	|	53.8		|	Breakout|		450.0|		ChopperCmd	|1150.0|
> |CrazyClimber|	9734.4	|	DemonAttack	|24074.1|		Freeway		|0.0|
> |Frostbite	|	249.7	|	Gopher		|1286.9	|	Hero			|8171.3|
> |Jamesbond	|525.0		|Kangaroo		|262.5	|	Krull		|7782.0|
> |KungFuMaster|	20543.8	|MsPacman	|1594.1		|Pong		|19.8|
> |PrivateEye	|96.9		|	Qbert	|	13651.6|		RoadRunner|	16809.4|
> |Seaquest	|	1215.6|		UpNDown	|12344.7|	|   |
>
> According to the original paper of IRIS, when the training data size is limited to 100k frames, the average score of IRIS is 1.046, which is lower than that of EfficientZero and GW-PCZero. We will add the results of IRIS to our paper in the revised version. The Dreamer in the original paper was trained with 200M frames, which is a very different setting from our paper. We will make every effort to report the result of the Dreamers under 100k training frames in the future.
> Q3: Can you report a performance with fixed coeff of PC loss?
> A3: We report the results below by fixing the coefficient at 0.35. The normalized mean score is 1.421, which is higher than EfficientZero’s 1.212 under the same training conditions. The value of 0.35 is a moderate setting for the coefficient of PC loss. In practice, we suggest to use a small coefficient for simple games, a large one for complex games. That is, the coefficient can be further adjusted for improved performance.
> |Game	|	Score	|	Game	|	Score	|	Game	|	Score	|
> | :-:   |   :-:             | :-:               |:-:   |   :-:             | :-:               |
> |Alien		|699.7		|Amidar		|160.1		|Assault		|1350.2|
> |Asterix	|	9000.0	|	BankHeist	|155.6	|	BattleZone	|9968.8|
> |Boxing		|27.4		|	Breakout	|	384.1|		ChopperCmd|	1150.0|
> |CrazyClimber|	6665.6	|	DemonAttack	|12116.6|		Freeway		|0.0|
> |Frostbite	|	249.7	|	Gopher	|	1769.4	|	Hero			|12646.4|
> |Jamesbond	|357.8		|Kangaroo	|	325.0	|	Krull		|7782.0|
> |KungFuMaster|	10400.0	|MsPacman	|805.0		|Pong	|6.7|
> |PrivateEye	|100.0		|Qbert		|7439.1		|RoadRunner|	5693.8|
> |Seaquest	|	1006.3	|	UpNDown	|3932.5	    |||
>
> Q4: Can you report the performance of GW-PCZero with 120k training steps?
> A4: The Path Consistency (PC) constraint is able to improve the model’s learning efficiency, and make it converge fast when the sample size is limited. It should be noted in this paper that the amount of game frames collected for training is fixed at 100k, regardless of whether the training steps are 60k or 120k. Therefore, increasing the number of training steps does not always lead to performance improvement, as GW-PCZero may converge early before the 120k step. In practice, we observe certain performance improvement in several games. For example, the score is improved from 19.8 to 20.6 for Pong, and the score is improved from 262.5 to 1793.8 for Kangaroo. The performances on most of the games remain the same roughly. Moreover, increasing the number of training steps from 60k to 120k, the training time is would be doubled. Also, we need to pay attention to appropriately adjusting  the decay rate of the learning rate in EfficientZero, because it is related to the number of training steps.  The comparison results between GW-PCZero against the full-version EfficientZero with 120k training steps on some of the Atari games are shown as follows. GW-PCZero has already converged in some games when training steps are 60k.The mean normalized score on those 10 games is 3.86 and 2.90 for GW-PCZero and EfficientZero accordingly. We will finish the computation on 26 Atari games in the revised version of the paper.
> |Game|GW-PCZero|EfficientZero|Game|GW-PCZero|EfficientZero|
> |:-:|:-:|:-:|:-:|:-:|:-:|
> |Breakout|450.0|406.5|DemonAttack|24074.1|13298.0|
> |Jamesbond|525.0|459.4|Krull|7782.0|6047.0|
> |MsPacman |1594.1 |1387.0|Kangaroo|1793.8|962.0|
> |Pong|20.6|20.6|Hero|10818.8 |8530.1|
> |Amidar|97.0|101.9|PrivateEye|96.9|100.0|

---

> ### Comment · Reviewer_s6tP · 2023-08-18
>
> 1. I would like to question the paper's novelty. It seems that your proposed GW-PCZero is very similar to Deepmind's' Self Consistent Models and Values' published in Neurips 2021. SCZero defines a sc-residual loss as $$l^{sc-residual}=\sum_{k=0}^K (\hat{r}(s_k, a_k)+\gamma \hat{v}(s_{k+1})-\hat{v}(s_k))$$, which is basically equal to your PC loss as $$l^{sc-residual}=\sum_{k=0}^K (g(s_{k+1})+\hat{v}(s_{k+1})-g(s_k)-\hat{v}(s_k))$$. It seems that your PC loss is just a multi-step version of SC loss, so I think the novelty of this article should be questioned unless you can prove that the effectiveness of PC loss is significantly improved compared to SC loss.
> 2. For A1, the median score should be a much higher value, such as EfficientZero's 1.09. This result cannot prove that you have a significant performance improvement compared to EfficientZero.
> 3. For A3, it seems that GW-PCZero's performance is extremely sensitive to the PC loss coeffiient. It seems that GW-PCZero cannot outperform EfficientZero with a fixed PC loss coefficient. I hope the author can release 120k training steps with fixed PC loss coeff.
>
> Considering the potential issues that may exist in the innovation of the article and the weakness of the experimental results, I will downgrade my score to 4.

---

> > ### Author Response · Authors · 2023-08-20
> >
> > Thank you for your response and for providing SCZero as a reference.
> >
> > Response to Q1:
> > Path consistency (PC), which is "f values on one optimal path should be identical", is rooted from the path optimality of the classical $A^*$ search algorithm (Hart et.al, 1968). __CNneim-A (Xu et al., 1987)__ relied on this optimality to use $A^*$ to make a lookahead scouting to estimate a segment on the optimal path and use the average of f-values from the root to the current state (i.e., the historical trajectory) and also one on this segment to guide $A^*$ search, and named this condition as path consistency. Subsequently, PC was suggested to cooperate with deep reinforcement learning to improve learning efficiency __(Xu, 2018)__. As shown in Equation (8) in __(Xu, 2018)__, PC was suggested to regularize the learning process by adding a weighted penalty to the loss function
> > $$
> > L(\theta)=\sum_{s\in Path}[w_sL_s(\theta)+w_cL_s^c(\theta)]+|\theta|^{\gamma_r},
> > $$
> > where $L_s(\theta)$ is the reinforcement learning loss that results from the interaction with the environment, $L_s^c(\theta)$ is the consistency loss, $w_s$ and $w_c$ are adjustable hyperparameters. $L_s^c(\theta)$ is evaluated as the deviation from an estimated value of optimal path, as suggested by Equation (9) in (Xu, 2018),
> > $$
> > L_s^c(\theta)=|f(s)-f^*(s)|^{\gamma}
> > $$
> > where $f^*(s)$ is a moving average of $f$ values of states within a segment window $W_s\subset Path$ with $s\in W_s$. If $\gamma$ is set as 2, $L_s^c$ will be $L_2$ deviation as used in this paper. Although The idea of using PC to improve the learning efficiency of reinforcement learning algorithms was already proposed (Xu, 2018), it needs further investigation to implement and test the potential of this direction. PCZero (Zhao et.al, 2022) had applied this idea to AlphaZero, demonstrating PC's effectiveness for the first time in board games. In this paper, we generalize the idea of PC from board games to scenarios where the environment emits immediate rewards, such as Atari games.
> > Path consistency is a __global constraint__ for all states on the optimal path. As illustrated in Equation (12), if the TD error for state $s_t$ with all states on the path is minimized to $0$, $s_t$'s PC loss $L_{PC}(s_t)$ is minimized. To facilitate practical implementation, the PC target is prepared within a selected window. But PC is still a global constraint on the entire path conceptually. SC loss is a local constraint associated with two adjacent states. For a given path, the TD error between states increases as the distance between them grows, under the SC loss constraint. Assuming the SC error of two adjacent states is $\delta$, the TD error between $s_t$ and $s_{t+k}$ might be amplified to $k\delta$, which is less reliable than PC loss. We replace the PC loss with SC loss to conduct experiments with 60k training steps in some of the Atari games. The mean normalized score of SC loss is 2.78, which is lower than PC loss's 3.76 but higher than EfficientZero's 2.21. __These experimental results indicate that SC loss can also improve learning efficiency, but PC loss is more effective than SC loss__.
> >
> > |Game|EfficientZero|SC loss|PC loss|Game|EfficientZero|SC loss|PC loss|
> > |:-:|:-:|:-:|:-:|:-:|:-:|:-:|:-:|
> > |Breakout|366.7|415.4|450.0|Qbert|8120.3|4540.6|13651.6|
> > |Assault|994.8|1114.2|1224.1|UpNDown|7592.5|14121.9|12344.7|
> > |Asterix|17734.4|18668.8|14771.9|CrazyClimber|8059.4|9953.1|9734.4|
> > |DemonAttack|7940.8|12743.9|24074.1|MsPacman|967.5|883.1|1594.1|
> > |Amidar|60.6|116.2|97.0|KungFuMaster|8956.3|8853.1|20543.8|
> > |Krull|3818.5|5147.9|7782.0|Score|2.21|2.78|__3.76__|
> >
> > __(Hart et.al, 1968)__ Hart, Peter E., Nils J. Nilsson, and Bertram Raphael. "A formal basis for the heuristic determination of minimum cost paths." IEEE transactions on Systems Science and Cybernetics 4.2 (1968): 100-107.
> > __(Xu et al., 1987)__ Xu, Lei, Pingfan Yan, and Tong Chang. "Algorithm cnneim-a and its mean complexity." Proc. of 2nd international conference on computers and applications. IEEE Press, Beijing. 1987.
> > __(Xu, 2018)__ Xu, Lei. "Deep bidirectional intelligence: AlphaZero, deep IA-search, deep IA-infer, and TPC causal learning." Applied Informatics. Vol. 5. No. 1. Berlin/Heidelberg: Springer Berlin Heidelberg, 2018.
> > __(Zhao, et.al, 2022)__ Zhao, Dengwei, Shikui Tu, and Lei Xu. "Efficient Learning for AlphaZero via Path Consistency." International Conference on Machine Learning. PMLR, 2022.

---

> > > ### Author Response · Authors · 2023-08-20
> > >
> > > Response to Q2:
> > > The median score reported in A1 is obtained with a training step of 60k. We will try our best to reproduce the results of EfficientZero and provide the results of our GW-PCZero when the training step is 120k.
> > >
> > > Response to Q3:
> > > The value of $\lambda$ is decided based on the game's complexity. In this paper, $\lambda$ is set to $0.2$ if the game is relatively simple. $\lambda$ is set to $0.35$ if the game is complex. In the following, we present the results of several simple games, for which the size of the action space is less than 18, with $\lambda=0.2$. The mean normalized score when $\lambda=0.2$ is 3.34, much larger than 2.29 when $\lambda=0.35$. For games with similar complexity, the value of $\lambda$ is not adjusted for different games. Simple games tend to exhibit better performance with smaller $\lambda$ values, while complex games tend to demonstrate better performance with larger $\lambda$ values.
> > >
> > > |Game|$\lambda=0.2$|$\lambda=0.35$|Game|$\lambda=0.2$|$\lambda=0.35$|
> > > |:-:|:-:|:-:|:-:|:-:|:-:|
> > > |Breakout|450.0|384.1|Pong|19.8|6.7|
> > > |Qbert|13651.6|7439.1|Assault|1224.1|1350.2|
> > > |UpNDown|12344.7|3932.5|Asterix|14771.9|9000.0|
> > > |CrazyClimber|9734.4|6665.6|DemonAttack|24074.1|12116.6|
> > > |MsPacman|1594.1|805.0|Amidar|97.0|160.1|
> > > |KungFuMaster|20543.8|10400.0|Score|3.34|2.29|

---

> > > ### Comment · Reviewer_s6tP · 2023-08-20
> > >
> > > I know the proposed story well and also have read about your previous ICML paper on AlphaZero. I believe this is not a deliberate 'coincidence'. However, although SCZero and GW-PCZero have totally different storyline, they are fundamentally talking about the same idea, which can be verified in the simple conversion between sc-loss and pc-loss. Therefore, this paper can hardly been proved as a innovation.
> > >
> > > Besides, although the proposed pc-loss can theoretically consider a much wider range even global one, it is a local-constraint the same as the sc-loss in implementation. But undeniably, the sc-loss can also gradually converge to the f-value consistency on a optimal path.
> > >
> > > I also have questions about your supplementary results of SC-loss. The SCZero proposed sc-direct and sc-reverse losses to prevent the potential modal collapse (all zero in reward and value predictions) of sc-residual loss. You didn't mention which one you used. Did you compare the difference between sc-direct and sc-residual loss?
> > >
> > > Overall, this paper introduces a interesting story about f-value consistency in nonzero immediate reward tasks, and did comprehensive studies based on Atari games. This further verifies the effectiveness of value consistency. But unfortunately, a extremely similar idea was proposed by DeepMind in Neurips 2021. However, this paper didn't cite or take SCZero as the most important baseline, nor make sufficient change, which led to its insufficient novelty. I thank the authors' work and response, but I cannot increase my score.

---

> > > > ### Author Response · Authors · 2023-08-21
> > > >
> > > > Thank you for your response and for providing us with the SCZero paper __(Farquhar et.al, 2021)__. We will include SCZero and other relevant works in our research paper as references and provide a comprehensive discussion to highlight the distinctions among these algorithms.
> > > >
> > > > Path consistency (PC) is a global constraint for all states on the optimal path. To facilitate practical implementation, the PC target is approximately prepared within a segment window. The more accurate the approximation, the better the performance. Due to time limitations, we have conducted experiments on some of the Atari games. The mean normalized score of SC direct residual loss is 2.78, which is lower than PC loss's 3.76 but higher than EfficientZero's 2.21, confirming that the current implementation of the PC loss contains more global information and thus is more effective than the SC loss. We will compare GW-PCZero with SCZero on more Atari games in future and discuss their similarities and differences in our paper.
> > > >
> > > > In Section 4.1 of our paper, we have proved that if TD relationships are established between any two states within the same optimal path, PC loss will achieve the optimal value. The connection between PC and TD is built for the first time in this paper. It should be noted that this conclusion is derived from a global perspective considering all states along the __entire path__. What's more, in this paper, we have proven that the neural-guided MCTS with path consistency as the constraint of the estimated values is guaranteed to find the optimal solution, when the number of simulations is sufficiently large. We have been conducting more theoretical research on path consistency and applying it to more algorithms and applications to demonstrate its effectiveness.
> > > >
> > > >
> > > > As displayed in Figure 1 of SCZero __(Farquhar et.al, 2021)__, Dyna __(Sutton et.al, 1991)__, VE learning __(Schrittwieser et.al, 2020)__, and SC-Direct are all trained by minimizing a TD objective. Specifically, the updating target used in Dyna __(Sutton et.al, 1991)__ is
> > > > $$
> > > > L^{Dyna}=r+\gamma\max_b Q(y,b)-Q(x,a),
> > > > $$
> > > > where $x$ is the starting state, $y$ is the next state, $a$ and $b$ are actions. MuZero __(Schrittwieser et.al, 2020)__ is a specific VE learning model, which is in the form of $k$ step TD error,
> > > >
> > > > $$
> > > > L^{VE}=[(\sum_{j=1}^k\gamma^{j-1}r_{t+j}+\gamma^k\hat{v}_{t+k})-\hat{v}_t]^2.
> > > > $$
> > > >
> > > > The SC loss __(Farquhar et.al, 2021)__ is also expressed as a form of TD error,
> > > > $$
> > > > L^{SC-residual}=[\hat{r}(s_t,a_t)+\gamma\hat{v}(s_{t+1})-\hat{v}(s_t)]^2.
> > > > $$
> > > >
> > > > Even though Dyna, VE learning, and SCZero all employ similar TD-like loss functions, there still exhibit some differences. As displayed in Figure 1 of SCZero, Dyna and SCZero use only trajectories generated from the model. The difference is that Dyna updates only the value predictions to be consistent with the model, but SC-Direct jointly updates both value and model to be self-consistent. VE learning is a grounded update that is similar in structure to SC, but uses real experience to compute the TD targets. Similar to the above discussions, __there are also differences between PCZero and SCZero from the above aspect__:
> > > > 1. SCZero is built on model-based algorithms, encouraging a learned model $\hat{m}$ and value function $\hat{v}$ to be jointly self-consistent, in the sense of jointly satisfying the Bellman equation with respect to $\hat{m}$ and $\hat{v}$. Path consistency requires that $f$ values on the same optimal path are consistent, which is suitable for value-based algorithms, which is not relative to whether the algorithm is model-based or not.
> > > > 2. In Dyna and SCZero, the reward $r$ and value $v$ in the TD target are provided by networks. Whereas in VE learning, both $r$ and $v$ are collected from real experience. PC is a value estimation constraint for states on the same optimal path, and reward $r$ is collected from real experience, and value $v$ is provided by the network.
> > > > 3. PC is evaluated by the deviation of $f$ value from its mean within a segment window, and a weighting mechanism is employed to provide a more reliable learning target.
> > > >
> > > > Dyna __(Sutton et.al, 1991)__, VE learning __(Schrittwieser et.al, 2020)__, SCZero __(Farquhar et.al, 2021)__  and PCZero are all related to TD error but exhibit distinct characteristics. We will further elaborate on their detailed comparison in the revision of the paper.
> > > >
> > > > __(Farquhar et.al, 2021)__ Farquhar, Greg, et al. "Self-consistent models and values." Advances in Neural Information Processing Systems 34 (2021): 1111-1125.
> > > > __(Sutton et.al, 1991)__ Sutton, Richard S. "Dyna, an integrated architecture for learning, planning, and reacting." ACM Sigart Bulletin 2.4 (1991): 160-163.
> > > > __(Schrittwieser et.al, 2020)__ Schrittwieser, Julian, et al. "Mastering atari, go, chess and shogi by planning with a learned model." Nature 588.7839 (2020): 604-609.

---

> > > > > ### Comment · Reviewer_s6tP · 2023-08-22
> > > > >
> > > > > Thanks for your response, but I have more questions now.
> > > > > 1. Obviously, the PC constraint is also compatible with the model-free methods. This is a special case of PC that only considers historical paths. But what's the benefit if using PC loss as an auxiliary loss? Can your provide any experimental results? You mentioned that using PC loss solely brought poor performance in the rebuttal Q4 to reviewer YYov.
> > > > > 2. If you take real rewards vs. predicted rewards as the major difference between PCZero and SCZero, how do you handle the off-policy issue? From the Eqn 12 and Figure 2 in PCZero(ICML), your proposed sliding window spans across historical path(real trajectory) and heuristic path(MCTS). However, since EfficientZero and MuZero are all off-policy algorithms, the historical path could be much 'older' than the heuristic path as you mentioned in Sec 4.3. In this regard, you proposed a weighting trick $w_i=c_b-c_a\cdot i$. This approach is essentially aimed at reducing the impact of 'far' rewards and values, both in heuristic path and historical path. Since the further the states are, the higher probabilities they are out of the on-policy path, no matter due to the off-policy-ness or the exploration noise. Therefore, the proposed PC loss cannot reach global constraints under the off-policy setting. Using the weighting trick seems no change compared to SCZero, which makes your novelty poor.
> > > > > 3. As reviewer YYov mentioned, path consistency holds not only the optimal path but also all on-policy paths. I think the best choice is combining PC loss with a on-policy model-based algorithms. In this way, you have no need to introduce the weighting trick and could use a much wider sliding window if the computation is accessible. The writing could also be simpler and more straightforward, just like the PCZero(ICML) paper. PCZero plays board games without immediate rewards, hence off-policy issues are not that serious. But undeniably, even in board games, PCZero's off policy issue still exist.
> > > > >
> > > > > Minor:
> > > > > 1. In Eqn 12, you should let the reviewer and audience know which reward is predicted and which reward is ground-truth.
> > > > > 2. Figure 1 right: you should mark the states with $s_{t-l},s^*_{t-l},\cdots,s_{t-1},s^*_{t-1},s_t$ to remind the readers this is a historical path illustration.

---

### Official Review · Reviewer_15wJ · 2023-07-06

**Soundness:** 3 good
**Presentation:** 3 good
**Contribution:** 2 fair
**Rating:** 5
**Confidence:** 3

**Summary:**

This paper proposes GW-PCZero, a reinforcement learning method, extending the technique of PCZero which is currently limited to board games and lacks theoretical backing. The GW-PCZero is designed for environments with non-zero immediate rewards, such as Atari games. It maintains path consistency by regularizing value estimation with the deviation from the mean value along the path, while a new weighting mechanism minimizes scouting variance. The paper provides the first theoretical proof that a neural-guided Monte Carlo Tree Search can guarantee finding an optimal solution under path consistency. And it reaches better performance on the Atari 100k benchmark with 26 games compared to the previous SoTA EfficientZero.

**Strengths:**

1. The paper is clear and understandable, especially in the details in Preliminary Section.
2. The paper provides a theoretical guarantee under the constraint of path consistency.
3. The performance increase in the experiments shown is laudable.

**Weaknesses:**

1. The novelty seems limited. Compared to PCZero, it modifies the PC target through a linear weighting method, which is a little tricky and common.
2. The authors claimed a theoretical guarantee for path consistency (PC) for the first time. But the PC is the main contribution of PCZero instead of GW-PCZero, which is confusing. However, there is no guarantee for the weighting method, concerning convergence rate or optimality.
3. No ablation for different weighting methods. Only the tradeoff c is considered. For example, as illustrated in Figure 1(left), why the weights are not exponentially decayed?

**Questions:**

1. As you mentioned in Definition 3.1 (L125), PC gives the constraint of the optimal path for the f values. For the optimal path, the f values keep the same in the path. But, I am wondering that "when f values match in the path, is this path optimal? especially when neural nets estimate the f values."

2. For tasks with much longer horizons, the computation cost of the path consistency can be much higher. How to reduce the corresponding cost in such cases?

**Limitations:**

1. The novelty of introducing a weighting mechanism without further clarification is limited. The authors claim that the weighting mechanism can mitigate the uncertainty/variance in L234, but no experiments or theorems are provided regarding the uncertainty/variance.

2. The experiments and analysis of the weighting mechanism are not enough, which is the main contribution of the work.

---

> ### Author Rebuttal · Authors · 2023-08-09
>
> We sincerely appreciate your valuable suggestions and would like to address the questions one by one as follows.
> Q1: The novelty seems limited. The authors claimed a theoretical guarantee for path consistency (PC) for the first time. But the PC is the main contribution of PCZero instead of GW-PCZero, which is confusing.
> A1: The primary contribution of this paper is to extend the concept of Path Consistency (PC) to a broader range of application scenarios with non-zero immediate reward, such as Atari games. In the literature, PC was considered by PCZero soly on board games, for which the immediate rewards are always zero. What’s more, PCZero only demonstrated the effectiveness of PC through experiments, and lacked theoretical guarantees. In this paper, we provide a theoretical foundation for PC for the first time that the neural-guided MCTS is guaranteed to find the optimal solution under the constraint of PC. Furthermore, we present a weighting mechanism into the calculation of the PC target to reduce the variance caused by scouting’s uncertainty and improve the performance.
> Q2: As you mentioned in Definition 3.1 (L125), PC gives the constraint of the optimal path for the f values. For the optimal path, the f values keep the same in the path. But I am wondering that "when f values match in the path, is this path optimal? especially when neural nets estimate the f values."
> A2: If the f values are estimated quite accurately, the path with the same f-value is the optimal path. The reason is that the root node is definitely on the optimal path, and the f value of the root node is the global optimal solution with highest reward. Therefore, the nodes which have the same f value as the root node also have the highest reward, and they constitute the optimal path. However, if the quality of the estimated f-values cannot be guaranteed, the above statement may not hold true. For example, one may construct a specific function that outputs the same f values for a randomly picked path which is unlikely to be optimal. In practice, such case should be seldom encountered, because the parameters of the neural network to predict the f value are usually randomly initialized.
> Q3: For tasks with much longer horizons, the computation cost of the path consistency can be much higher. How to reduce the corresponding cost in such cases?
> A3: The computational complexity of PC mainly lies in preparing the learning target, which computes the mean f value for all states within a selected window. If the task has much longer horizons, we can still compute the PC target based on the predefined k nearest neighbor nodes. The number k is a given constant, and it is irrelevant to the length of the horizons. According to our empirical experience, k=5 is good enough.
> Q4: The authors claim that the weighting mechanism can mitigate the uncertainty/variance in L234, but no experiments or theorems are provided regarding the uncertainty/variance.
> A4: For a sampled batch ${s_0,s_1,s_2,s_3}$, the PC target is calculated as
> $$\bar{f}=\frac{f(s_0 )+w_1 f(s_1 )+w_2 f(s_2 )+w_3 f(s_3 )}{1+w_1+w_2+w_3}$$
> As mentioned in the paper, PC requires that f values of states along any optimal path in a search graph should be identical with $s_0$, and the probability that $s_i$ and $s_0$ are not in the same optimal path grows as i increases. If we assume the f value follows a normal distribution,  $f(s_i)$ has the same mean value but the variance grows as i increases, because the probability of $f(s_i)$ being farther away from the mean grows. When a smaller weight $w_i$ is given to $f(s_i)$ with a larger index i, the variance of the estimated $\bar{f}$ will be reduced. From this perspective, the weighting mechanism is a reasonable way to mitigate the variance.
> Q5: More ablation study on the weighting mechanism.
> A5: Thanks for your suggestion. We provide experimental results of the exponential weighting approach, for which the weight is calculated as $\exp⁡\\{-i×0.1\\}$. Due to the limited time of the rebuttal period, we report the results on some Atari games as follows. Notice that the performance is improved on seven out of twelvegames if replacing the linear weighting with the exponential weighting method. It deserves more investigations on the weighting mechanism in the future.
> |Game	|	Linear	|	Exp	|	Game	|	Linear	|	Exp	|
> | :-:   |   :-:             | :-:               |:-:   |   :-:             | :-:               |
> |Breakout	|	450.0|	475.7	|	Pong		|19.8	|	20.2|
> |Qbert		|13651.6|	11737.5	|	Assault		|1224.1	|1224.9|
> |Asterix	|	14771.9|	15750.0|		CrazyClimber|	9734.4|	6718.8|
> |MsPacman	|1594.1	|1319.7		|Amidar	|	97.0		|186.2|
> |KungFuMaster|	20543.8|	24025.0|	Krull	|	7782.0|	6131.0|
> |Alien		|699.7	|627.8|		Frostbite	|	249.7|	258.1|

---

> ### Comment · Reviewer_15wJ · 2023-08-20
>
> Thank you for your reply and experiments. But considering the novelty (Reviewer s6tP also mentioned SCZero, which is quite similar to this work), I will keep my score.

---

### Official Review · Reviewer_YYov · 2023-07-08

**Soundness:** 2 fair
**Presentation:** 3 good
**Contribution:** 2 fair
**Rating:** 4
**Confidence:** 4

**Summary:**

This paper extends PCZero to more general games whre the environment emits non-zero immediate rewards and proposes Generalized Weighted PCZero (GW-PCZero). GW-PCZero is built on EfficientZero with a generalized PC constraint. Specifically, GW-PCZero add an additional value consistence loss alone the sampled path, i.e., $L_{PC}(\theta) = \left(v(s_t;\theta)-\frac{1}{l+1} \sum_{i=0}^l\left[\sum_{j=1}^i r_{t+j}+v\left(s_{t+i} ; \theta\right)\right]\right)^2$, the MSE loss between $v(s_t;\theta)$ and the average value of {$0$-step td-target, $1$-step td-target, ..., $l$-step td-target}. To reduce the bias and variance of the PC targets when training with off-policy data, GW-PCZero devise a weighting mechanism to give larger discounts to farther states, which is very similar to the idea of `td-lambda`. So, this paper can be considered as adding an additional `td-lambda` style value loss to EfficientZero. Experiments on the Atari-100k benchmark validate the superiority of GW-PCZero over EfficientZero.

**Strengths:**

* The paper is written clearly.
* The proposed GW-PCZero becomes a new SOTA on the Atari-100k benchmark, which achieves a higher human normalized score than EfficientZero but with much less computational cost.

**Weaknesses:**

* 1. The path consistency exists not only on the `optimal` path but on any `on-policy` path.
* 2. Although the paper has done a lot of theoretical analysis, the proposed method can be considered as  adding an additional `td-lambda` style value loss to EfficientZero. Therefore, the differences between the new proposed value loss functions in Eq. (15) and (17) with other typical value loss functions, e.g., td-lambda, v-trace, etc, should be discussed.
* 3. For EfficientZero, the policy target reanalyze process and the value target reanalyze process are not necessarily separated into 2 independent passes.
    * We can sample a slightly longer sub-trajectory $L_b=\\{s_t, s_{t+1}, \cdots, s_{t+H}\\}$ for each item in a batch (but keep the total transition number in a batch the same), where $H>l$ and use a single batch of MCTS to get $\pi^{MCTS}$ and $v^{MCTS}$ for all states in $L_b$. To compute $z_t, \ldots, z_{t+H-l}$, we follow Eq. (9) in this paper but replacing $v$ with $v^{MCTS}$. For $z_{t+H-l-1}, \ldots, z_{t+H}$, we set the $n$-step td targets with smaller $n \in \\{l-1,\ldots, 0\\}$, e.g., we set $z_{t+H}=v^{MCTS}(s_{t+H})$.
    * Therefore, the computational cost (the number of MCTS runs) of EfficientZero is not necessarily much greater than GW-PCZero.
* Minors:
  * $v(s_{t+1};\theta)$  (line 146) should be $v(s_{t+l};\theta)$.
  * $L_p$ (line 221) should be $L_b$.


**Questions:**

* What's the performance of GW-PCZero if setting the updating steps to 120k?
* What's the performance of EfficientZero if directly using Eq. (9) as the value target (do not replace $v(s_{t+l};\theta)$ with $v^{MCTS}(s_{t+l};\theta)$?
* For $L_b=\\{s_t, s_{t+1}, \cdots, s_{t+H}\\}$, only $s_t$ is considered to be constrained with PC loss in real implementation because the sampled batch $L_p$ is too short to deal with the subsequent states in the same way (line 220-221). Why not directly using Eq. (12) as the loss function (which can utilizing all states in $L_b$)?
* What if removing the value loss (of EfficientZero) in GW-PCZero and only keeping the PC-constant loss shown in Eq. (12)?
* What if removing the PC-constant loss of GW-PCZero and replacing the value loss (of EfficientZero) with a `td-lambda` style or a `v-trace` style value loss?

**Limitations:**

Please see the weaknesses and questions.

---

> ### Author Rebuttal · Authors · 2023-08-10
>
> We sincerely appreciate your valuable suggestions and would like to take this opportunity to address the raised issues.
> Q1: What's the performance of GW-PCZero if setting the updating steps to 120k?
> A1: The Path Consistency (PC) constraint is able to improve the model’s learning efficiency, and make it converge fast. The amount of game frames collected for training is 100k, regardless of whether the training steps are 60k or 120k. Increasing the number of training steps does not always lead to performance improvement. In practice, we observe certain performance improvement in several games. For example, the score is improved from 19.8 to 20.6 for Pong, and from 262.5 to 1793.8 for Kangaroo. Increasing the number of training steps 120k doubles the training time. The comparison results between GW-PCZero against the full-version EfficientZero with 120k training steps are shown as follows. GW-PCZero has converged in some games after 60k updates.The mean normalized score on those 10 games is 3.86 and 2.90 for GW-PCZero and EfficientZero accordingly.
> Game|GW-PCZero|EfficientZero|Game|GW-PCZero|EfficientZero
> -|-|-|-|-|-
> Breakout|450.0|406.5|DemonAttack|24074.1|13298.0
> Jamesbond|525.0|459.4|Krull|7782.0|6047.0
> MsPacman |1594.1 |1387.0|Kangaroo|1793.8|962.0
> Pong|20.6|20.6|Hero|10818.8 |8530.1
> Amidar|97.0|101.9|PrivateEye|96.9|100.0
>
> Q2: What's the performance of EfficientZero if directly using Eq. (9) as the value target?
> A2: The comparison results with or without MCTS in Eq. (9) is shown in Table (9) in the appendix of EfficientZero. In this paper, the results of EfficientZero in Table 2 were obtained by adopting Eq (9) as the value target. We also add an experiment to implement EfficientZero to use MCTS root value correction. 5 games have performance improvement, while the other 5 are not.
> Game|With MCTS	|Without MCTS|Game|With MCTS|Without MCTS
> -|-|-|-|-|-
> Alien|638.8|850.6|Amidar|80.1|60.6
> Assault|	1352.5|994.8| Asterix|19356.3|17734.4
> BankHeist|293.8|276.9|BattleZone|13718.8|15875.0
> Boxing|43.3|28.2| Breakout	|357.2|366.7
> ChopperCmd|631.3|818.8|CrazyClimber|7115.6|8059.4
>
> Q3: Why not directly using Eq. (12) as the loss function?
> A3: In practice, it is usually not a good choice to use Eq. (12) that require all nodes along the entire path to be available. First, the entire path may be long with a large number of nodes, and the computation for the PC target would be of high complexity. Secondly, obtaining complete and terminated paths might not be feasible in situations like Atari. Third, as mentioned in the paper, using the entire path to prepare the PC target may be unreliable. In practice, we suggest to select a certain number of the neighboring nodes through a weighting mechanism. This idea of local computation of Eq. (12) is left for future.
> Q4: What if removing the value loss and only keeping the PC loss?
> A5: The results of the PC-soly version are worse than the version where both loss functions are considered. Same as the PCZero paper, value loss and PC loss cannot be replaced withby each other, and the better performance will be achieved if both are adopted.
> Game|Random Player|PC-soly|PC + Value
> -|-|-|-
> Qbert|163.9|2138.3|13651.6
> Assault|222.4|229.7|1224.1
> Asterix|210.0|2675.0|14771.9
> DemonAttack|152.1|3947.0|24074.1
> MsPacman|307.3|1186.3|1594.1
>
> Q5: The differences between the PC loss with other typical value loss functions should be discussed. What if removing the PC loss and replacing the value loss with a td-lambda style or a v-trace style value loss?
> A5: The path consistency loss and other typical value loss such as TD-lambda share many similarities. The PC target in Eq. (17) is to combine all i-step returns through a weighting mechanism, and it contains TD-lambda as a special case if we set the weights to $(1-\lambda)\lambda^{i-1}$. Other weighting methods can be considered for PC. PC loss and TD-lambda have some differences. In general, learning with the PC is more flexible than TD-lambda, because PC can be implemented in various mathematical forms. While preparing the learning target for PC, both states after $s_t$ and states before $s_t$ can be considered, and TD-lambda only considers the TD relationships with the states after $s_t$. If removing the PC loss of GW-PCZero and replacing the value loss with a td-lambda loss, we report the scores of several Atari games as and the performance is poor. If we keep the value loss and replace the PC loss with TD loss, the algorithm still works. TD-lambda is a special case of PC loss, and can be used as a substitute for PC loss. Both PC loss and TD-lambda cannot replace the role of the value loss.
> Game|Without value loss|With value loss|Game|Without value loss|With value loss
> -|-|-|-|-|-
> Breakout|3.22|415.4|Asterix |275.0| 18668.0
> MsPacman|606.9|883.1|Amidar|2.0|116.2
> Krull|1982.0|5147.9|Alien|577.8|916.9
>
> Q6: The computational cost of EfficientZero is not necessarily much greater than GW-PCZero.
> A6: Theoretically, the policy target reanalyze process and the value target reanalyze process can be done simultaneously if H is much larger than L, i.e., H >> L, and the reliability of most value function estimations is comparable to that by performing two MCTS runs. However, in practical situations, H is usually small, because a large H indicates long sub-trajectories and it tends to make the batch data violate the independence of training samples. For both MuZero and EfficientZero, H was 5. L was also 5 for MuZero. A dynamic L was adopted in EfficientZero, which is 4 or 5 in most cases. If H=L or H is slightly larger than L, a significant proportion of the value target is still determined by the last state, imposing higher requirements on the reliability of the value provided by MCTS. Therefore, EfficientZero need to employ two times of MCTS to ensure its performance. What’s more, PC can improve the learning efficiency greatly, reducing the computational resources consumption to half with 60k training steps.

---

> > ### Comment · Reviewer_YYov · 2023-08-22
> > **Response to Rebuttal**
> >
> > Thank you for your reply and experiments. After reading the comments of the other reviewers, I still have some concerns.
> >
> > (1) The authors did not replay to Weaknesses 1, which is the main concern: ``` the path consistency exists not only on the optimal path but on any on-policy path``` (Reviewer ```15wJ``` and ```s6tP``` also have similar questions). So, whether adding the path consistency constraint throughout the whole training cycle will make training less effective (especially when using too much off-policy data)? Whether the model is more likely to fall into suboptimal solutions?
> >   * From the authors' response (A2) to Reviewer ```15wJ```, I found the authors hold the viewpoint that when f values match in the path, the path is optimal, which I think is not correct.
> >
> > (2) After reading the comments of the other reviewers, I agree that the novelty may be limited compared to PCZero.
> >
> > So, I keep the score currently.

---

### Decision · Program_Chairs · 2023-09-21

**Decision:**

Accept (poster)

**Comment:**

Based on the provided reviews, I recommend accepting the paper for NeurIPS. The proposed Generalized Weighted PCZero (GW-PCZero) demonstrates strong contributions by extending path consistency to Atari games, introducing a weighting mechanism to enhance learning targets, and providing theoretical grounding for neural-guided Monte Carlo Tree Search under path consistency. The experiments showcase competitive performance gains over state-of-the-art methods while utilizing significantly fewer computational resources. Although the method draws from prior works, it still represents a valuable combination leading to improved results. The reviews reflect positive evaluations and strong contributions.